# The Influence of Cohesin on the Short-Scale Dynamics of Intact and Damaged Chromatin in Different Phases of the Cell Cycle

**DOI:** 10.3390/ijms26188837

**Published:** 2025-09-11

**Authors:** Vladimir S. Viushkov, Nikolai A. Lomov, Polina O. Kalitina, Daria M. Potashnikova, Anastasia S. Shtompel, Sergey V. Ulianov, Sergey V. Razin, Mikhail A. Rubtsov

**Affiliations:** 1Department of Molecular Biology, Faculty of Biology, Lomonosov Moscow State University, 119234 Moscow, Russiama_rubtsov@mail.ru (M.A.R.); 2Institute of Gene Biology of the Russian Academy of Sciences, 119334 Moscow, Russia; 3Department of Cell Biology and Histology, Faculty of Biology, Lomonosov Moscow State University, 119234 Moscow, Russia

**Keywords:** CRISPR-Sirius, RAD21, cohesin, auxin-inducible degron, chromatin dynamics, double-strand breaks repair, live-cell imaging

## Abstract

Cohesin organizes the genome into spatially segregated loops and topologically associated domains by loop extrusion. In addition, it ensures cohesion of sister chromatids after replication. Thus, cohesin is expected to limit chromatin dynamics by ensuring cohesion and compacting chromatin in the interphase. Nonetheless, loop extrusion is an example of chromatin dynamics; thus, cohesin could promote the dynamics of genomic loci at the scale of individual loops and contact domains. Moreover, given that the extruding activity of cohesin after replication is supplemented by its cohesive activity, the impact of cohesin on chromatin dynamics in different phases of the cell cycle may vary. Of particular interest is the cohesin’s role in the regulation of the dynamics of damaged chromatin, which remains insufficiently studied. Here, we visualized a genomic locus using the CRISPR-Sirius system in human cells with auxin-induced depletion of the cohesin subunit RAD21. Cohesin depletion increased the local spatial dynamics of the visualized locus on a time scale of fractions of a second to one minute. This effect was observed in both replicated and unreplicated chromatin. However, the increase in the mobility of the visualized locus upon cohesin depletion was more pronounced in the former. In addition, we showed that cohesin depletion did not affect the local mobility of double-strand break repair foci visualized using a fluorescent fragment of the repair factor 53BP1. Cohesin depletion did not affect the local mobility of repair foci in either replicated or unreplicated chromatin. The results indicate that cohesin constrains local spatial dynamics of genomic loci. At the same time, cohesive activity of cohesin is not indispensable for restricting chromatin dynamics, although it enhances the confinement effect. On the other hand, repair foci are less mobile structures than point chromatin loci, and cohesin does not affect their dynamics on the studied time scales.

## 1. Introduction

Cohesin is a large ring-shaped protein complex that consists of SMC1, SMC3, RAD21, and several additional subunits (reviewed in [1,2,3]). The canonical function of cohesin is to maintain sister chromatid cohesion from the S phase of the cell cycle until the anaphase of mitosis [2,4,5,6]. Moreover, it has been convincingly shown in the last decade that cohesin also organizes chromatin architecture into loops and loop domains throughout the interphase via loop extrusion [1,2,7,8,9,10,11]. Thus, cohesin could have a dual influence on short-scale spatial dynamics (i.e., local chromatin mobility). On the one hand, the organization of chromatin into contact domains (i.e., chromatin compaction) may limit chromatin dynamics. This constraining effect may be enhanced in the S phase of the cell cycle, when chromatid cohesion is added. On the other hand, loop extrusion by cohesin represents chromatin dynamics per se. According to theoretical models of loop extrusion, topological domain formation is in a nonequilibrium state: cohesin complexes are constantly loaded onto chromatin and form multiple transient loops, and also periodically dissociate from chromatin [12,13,14]. Hence, on the scale of individual loops within contact domains, cohesin may promote local chromatin dynamics by transiently bringing loci within a loop closer together. Research has shown that even extended cohesin-dependent loops detected in population Hi-C maps and by microscopy are not stable but short-lived and dynamic, with the lifetime of the fully looped conformation encompassing several minutes [15,16]. Moreover, most of the loop anchors are in a partially extruded or fully unlooped state [15,16]. This observation suggests that although cohesin activity compacts chromatin into topological domains observed in the analysis of cell populations, chromatin within topological domains can be quite dynamic at the level of individual cells, also due to the activity of cohesin itself. Considering that the cohesive activity of cohesin appears in the S-phase in addition to its extruding activity, the impact of this protein on the dynamics of unreplicated and replicated chromatin may differ and requires further study.

In addition to its key function in maintaining genome topology, cohesin participates in the repair of DNA double-strand breaks (DSBs) [17,18,19,20]. The role of cohesin in repair is manifested mainly in the S and G2 phases of the cell cycle [18,21], but there is evidence of a decrease in repair efficiency upon knockdown of the SA2 subunit of cohesin in human cells in the G1 phase (i.e., before replication and cohesion establishment) [19]. The importance of cohesin for preserving genomic integrity is also supported by the fact that mutations in the cohesin subunit genes are often found in tumor cells and may be one of the causes of genomic instability in such cells [19,22,23,24]. Knockdown of cohesin subunits in human cell culture increased the number of chromosomal rearrangements, emphasizing the role of cohesin in DNA break repair and genomic stability [20,21]. It has been suggested that the function of cohesin in repair may be to decrease the mobility of DSBs. Given that DNA repair impairments are more pronounced during the S and G2 phases of the cell cycle, chromatid cohesion was thought to be behind the effect of cohesin in repair [18,21,25]. However, some studies suggest more specific functions of cohesin in repair, such as participation in the formation of γH2AX repair domains through loop extrusion [26] and repression of transcription at the site of damage, before and after replication [19]. Additionally, there is some evidence that cohesin is required for activation of the cell cycle arrest upon DSB introduction [27,28,29].

Here, we aimed to investigate the effect of cohesin depletion on the spatial dynamics of a genomic locus, considering the impact on replicated and unreplicated chromatin separately. For this purpose, we generated HCT116 cells with a system for auxin-induced depletion of the RAD21 subunit of the cohesin complex. We used the CRISPR-Sirius system for chromatin visualization [30,31,32]. This system is based on catalytically inactive Cas9 (dCas9), a guide RNA that directs the system to the target locus and contains eight MS2 repeats and the MCP recognizing such repeats. In our version of the CRISPR-Sirius system, the MCP is fused with the superfolder GFP fluorescent protein (sfGFP). We have previously shown that this version of the system can be used to visualize chromatin loci in HCT116 cells [33]. In contrast to widely used imaging systems based on bacterial operators and repressors (e.g., LacO/R or TetO/R), the CRISPR-Sirius system requires fewer repeats: >20 according to our data [33] and the authors of the original technology [30], versus >100 repeats for repressor-operator systems [15,16,34,35]. In addition, the CRISPR-Sirius system uses endogenous genomic repeats rather than artificially integrated ones, allowing us to study chromatin dynamics in a native context. Using confocal time-lapse microscopy with an imaging step of ~0.4 s and determining the biophysical parameters of the trajectories, we demonstrated that cohesin (RAD21 subunit) depletion increases the mobility of the visualized genomic locus in replicated and unreplicated chromatin.

We also studied the effect of cohesin depletion on the spatial dynamics of repair foci visualized using the fluorescent reporter BP1-2 upon induction of double-strand DNA breaks by the topoisomerase poison etoposide. This reporter is the foci-forming region of the DSB repair factor 53BP1 [36]. However, under the same imaging time scale, we did not detect changes in the mobility of the BP1-2 foci upon cohesin depletion, either in G1 cells or S/G2 cells. This suggests that repair foci in human cells are fairly stable structures, and their spatial stability does not depend on cohesin on the studied time scales (subseconds to one minute).

## 2. Results

### 2.1. Cell Line with Auxin-Induced Degradation of RAD21 and with the CRISPR-Sirius Imaging System

Since knockout of cohesin genes is lethal [37,38], cells with inducible rapid cohesin depletion are needed to study the role of cohesin in the modulation of the spatial dynamics of chromatin. The auxin-inducible degron system has been successfully used for rapid cohesin depletion previously [7,8,39,40]. This approach allows for fast synchronous degradation of the target protein in cells, minimizing the risk of secondary effects that may occur with the long-term suppression of gene expression by RNA interference. In the auxin-inducible degron (AID) system, an amino acid sequence called degron (mAID in the chosen version of the system) is added to the protein, and the cells express the F-box subunit of the ubiquitin ligase TIR1 from *Oryza sativa* (OsTIR1) [39,40]. OsTIR1 polyubiquitinates the tagged protein upon auxin addition, targeting it to proteasomal degradation (Figure 1A).

As in previous studies, we used the RAD21 subunit of the cohesin complex as a target [7,8,15,41]. We integrated the mAID degron into the RAD21 gene in HCT116 cells, using CRISPR-mediated knock-in, adding the degron to the C-terminus of the RAD21 protein. The cells were cloned, and the cell clone for which homozygous integration was confirmed by PCR (Appendix A) was transduced with a lentiviral vector with the OsTIR1 ubiquitin ligase gene. The transduced cells were cloned, and one of the clones with the integration of the OsTIR1 gene was selected for further work. This cell clone is referred to as HCT116_RAD21_AID throughout the text. We confirmed the depletion of the target protein (RAD21) upon addition of auxins—a mixture of sodium salts of α-naphthaleneacetic (α-NAA) and indoleacetic acids (IAA)—using Western blotting (Figure 1B, see also Appendix A). The degree of RAD21 depletion in HCT116_RAD21_AID cells, assessed by Western blot, was >85% after 4–6 h of cell incubation with auxins. The half-degradation time was under 1 h. We performed ChIP-seq analysis with antibodies to RAD21 to verify that auxin addition lowers the amount of chromatin-bound cohesin in HCT116_RAD21_AID cells. We found that auxin addition results in the almost complete disappearance of cohesin peaks (Figure 1C,D). Meanwhile, cohesin-binding peaks in cells with the AID system, even in the absence of auxin, were smaller than in wild-type HCT116 cells. This result suggests basal degradation, which is described in the literature for the chosen variant of the auxin degron system [8,16,39,42,43,44,45]. Nonetheless, despite the reduced ChIP signal intensity, auxin-induced RAD21 depletion is evident in HCT116_RAD21_AID cells.

We examined the cell cycle profile upon incubation of cells with auxins and analyzed the growth curves to further characterize the HCT116_RAD21_AID cell line. Incubation of HCT116_RAD21_AID cells with auxins resulted in a cell cycle block in G2/M (Figure 1E, 12 h of incubation, see also Appendix A), the expected effect of cohesin depletion [15,41,46]. Interestingly, we also found a slight increase in the proportion of cells in the S phase upon RAD21 depletion. However, auxins did not influence cell cycle progression in control HCT116 cells (Figure 1E). In the absence of auxins, HCT116_RAD21_AID cells proliferated similarly to control HCT116 cells. Incubation with auxins suppressed the proliferation of HCT116_RAD21_AID cells but not of control HCT116 cells, which is consistent with the requirement of cohesin for normal mitotic progression (Figure 1F). Given that HCT116_RAD21_AID cell proliferation and cell cycle phase distribution did not differ significantly from wild-type HCT116 cells, possible basal degradation of RAD21 seems non-critical.

We determined the mitotic index of HCT116_RAD21_AID cells upon incubation with auxins to additionally characterize the proliferative defects. The mitotic index increased from approximately 5% to 16% upon incubation of cells with auxins for 12 h, confirming that mitosis was blocked (Figure 2). We also found that upon cohesin depletion, cells with abnormal nuclear morphology accumulated in the culture, reaching >40% by 24 h of incubation with auxins (Figure 2D). The abnormal nuclear morphology was manifested as multilobulated, highly segmented nuclei and nuclei connected by thin chromatin bridges (Figure 2B,C). Such nuclei have been previously shown to form via endomitosis induced by cohesin depletion [41]. It can be proposed that the disruption of chromosome segregation in such endomitotic cells may be due to chromosome fusion resulting from DSB misrepair that occurs following cohesin depletion [21]. In summary, the phenotypes observed upon incubation with auxins indicate that we achieved functional cohesin depletion in HCT116_RAD21_AID cells, reproducing the phenotypes described in the literature [7,15,16,41,46].

We used the CRISPR-Sirius technology for chromatin visualization (Figure 3A [30,31,32,33]). This technology is based on the binding of catalytically inactive Cas9 (dCas9) to a target locus to which it is directed by a guide RNA. An array of eight MS2 bacteriophage hairpins is added to the tetraloop of the guide RNA in the Sirius system. These hairpins are bound by the MS2 phage coat protein (MCP) fused with the superfolder GFP fluorescent protein (MCP-sfGFP). For visualization, the target locus should contain a cluster of locus-specific repeats to which the guide RNA is directed; this is necessary to achieve sufficient signal brightness. In a previous study, we showed that CRISPR-Sirius technology can be used to visualize chromatin loci associated with cohesin-binding sites in HCT116 cells [33]. Here, we chose one of these loci (which we named C6) as a model system. This locus is located in the third intron of the *TMEM242* gene on chromosome 6 and contains 61 repeats for the two gRNAs used for visualization. The cluster of visualized repeats in the C6 locus is adjacent to the cohesin peak, suggesting that the spatial dynamics of this locus may depend on cohesin activity (Figure 3B). The *TMEM242* gene encodes a mitochondrial transmembrane protein involved in the assembly of ATP synthase [47]. In HCT116 cells, *TMEM242* is expressed at a low level—approximately 1% of the housekeeping gene *GAPDH*, according to PRO-seq data from [7]. Its expression is not affected by cohesin depletion, as previously shown in HCT116 cells with a similar auxin degron system (according to analysis of DESeq2 data from [7]). Using lentiviral transduction, we integrated the genes encoding dCas9, MCP-sfGFP, and two guide RNAs targeting the C6 locus into the genome of HCT116_RAD21_AID cells, allowing us to visualize this locus in the cells (Figure 3C–E). Thus, we developed a model system for studying the role of cohesin in the regulation of a chromatin locus dynamics—cells with a system of auxin-induced depletion of the RAD21 cohesin subunit and visualization of the C6 genomic locus.

### 2.2. Cohesin Depletion Increases the Spatial Dynamics of the Visualized Locus

Using the obtained model cell culture, we studied the effect of RAD21 depletion on the spatial dynamics of the visualized locus. The dynamics of chromatin loci upon cohesin depletion were previously studied on fairly large time scales (from 10 s to 30 min), using a different visualization system (TetO/R system) [15]. Thus, we decided to look at the effect of cohesin depletion on chromatin dynamics on shorter time scales to study more local dynamics. For this purpose, we analyzed confocal time-lapse series of cell images (Figure 4A, see also Appendix A) with a time step of ~0.4 s. First, for each series, we obtained a track, a set of signal coordinates in time. Each track point was translated into the coordinates of the nucleus center in the corresponding frame to compensate for the motility of the entire nucleus. We considered only one signal in each cell (the best recognized one or randomly selected in the case of the same brightness/quality of recognition) to avoid pseudoreplication when constructing samples. To quantify the motility, we calculated the mean displacement per frame from the trajectory, reflecting the speed of the signal movement, and the gyration radius of the trajectory (also known as the length of constraint, Lc), reflecting the confinement of the trajectory [31,32,48,49]. We also investigated the trajectories using mean square displacement (MSD) analysis (reviewed in [49,50]). For this, a series of mean squared displacements of trajectory points for different time lags was calculated, and MSD curves reflecting the dependence of MSD values on time lags were plotted. The more mobile the visualized locus, the higher its MSD curve goes. Fitting individual curves with the anomalous diffusion model allowed us to obtain the motion parameters: the apparent (generalized) diffusion coefficient (D_App_) and anomalous exponent (α), the latter reflecting the degree of movement confinement [31,32].

We first examined the dynamics in unsynchronized cells. In untreated cells, the C6 locus exhibited subdiffusive dynamics (Figure 4B,C). Incubation of cells with auxins elevated the MSD curve of the C6 locus, reflecting an increase in the mobility of this locus upon cohesin depletion (Figure 4B). By contrast, auxin treatment of control HCT116 cells with visualized C6, but without the degron system, did not change the mobility of the visualized locus, suggesting the absence of a nonspecific effect of auxins on chromatin mobility. RAD21 depletion enhanced all analyzed movement parameters (Figure 4C): the anomalous exponent (α) increased by 1.37 times; the diffusion coefficient, by 1.50 times; the mean displacement, by 1.23 times; and the gyration radius, by 1.36 times (the ratio of sample medians is indicated in all cases). This outcome means that cohesin depletion increases the movement speed of the visualized chromatin locus (estimated by the mean displacement and diffusion coefficient) and the space available for diffusion (estimated by the anomalous exponent and the gyration radius). Meanwhile, the anomalous exponent remained below 1 (a median value of 0.31 in treated cells vs. 0.23 in untreated cells). This result indicates that factors limiting chromatin diffusion are preserved, even with cohesin depletion. This is expected for a cell nucleus, which is rich in long twisted DNA molecules, chromatin proteins, and nuclear subcompartments.

### 2.3. Cohesin Constrains the Dynamics of Both Replicated and Unreplicated Chromatin

The functions of cohesin vary throughout the cell cycle: it mediates loop domain formation via extrusion throughout the interphase and mediates sister chromatid cohesion from the S phase until mitosis. Thus, the effect of cohesin on replicated and unreplicated chromatin dynamics may differ, and this difference may be masked when chromatin dynamics are studied in an unsynchronized cell population. In addition, we noticed that auxin treatment for 5 h induced a decrease in the proportion of cells in the G1 phase (from 72% to 61%) and a slight increase in cells in the S, G2, and M phases (Figure 5A, “Standard cultivation”, see also Appendix A). This is a manifestation of the previously discovered cell cycle block, which is more pronounced with longer incubation (see Figure 1E, 12 h of incubation with auxins). Changes in the ratio of cells in different phases of the cell cycle can lead to biases in the assessment of changes in chromatin mobility upon cohesin depletion. Therefore, we studied the effect of cohesin depletion in cells with replicated and unreplicated C6 locus separately. We used the above-described sample of unsynchronized cells to select the cells in which the signals were doubled to compile a sample of cells with replicated C6 locus (Figure 5B). On the other hand, cells from the unsynchronized population, in which no signal doubling was observed, cannot be confidently considered G1 cells because signal doubling may not be noticeable as a result of diffraction limitations of optical microscopy. In addition, the visualized locus may not have replicated yet at the time of microscopy in some cells in the S phase. Thus, we cannot classify cells with non-doubling signals as G1 cells (see, for example, the cell in Figure 5B, in which one of the signals is not replicated). We noticed, however, that if the cells were incubated on a glass dish not for 1 day before microscopy (the standard condition in our work), but for 5 days, the cells accumulated in the G1 phase (up to 85%), probably because of contact inhibition (Figure 5A, “Prolonged cultivation”, see also Appendix A). Moreover, when auxins were added, such cells did not leave the G1 state, suggesting they stopped in the G1 phase of the cell cycle. We compiled samples of signals visualized in such cells that corresponded to signals in the G1 phase of the cell cycle in auxin-treated and control cells.

The analysis of cells with replicated signals showed that the motility of the visualized signals increased upon cohesin depletion. This was manifested by an enhanced MSD curve and elevated mobility parameters (Figure 5C). Thus, cohesin restricts the dynamics of the visualized locus in replicated chromatin. Interestingly, we also found an increase in motility in non-replicated chromatin in G1 cells (Figure 5D). This observation implies that cohesion is not indispensable for the restriction of chromatin dynamics by cohesin. Thus, local chromatin dynamics are restricted by cohesin not only because of cohesion establishment, but probably also because of the protein’s activity in loop domain formation. However, cohesin depletion in G1-phase cells resulted in a less pronounced change in mobility compared to S/G2 cells. We did not find a significant change in the anomalous exponent in the former, and the increase in other parameters was also less pronounced than in S/G2 cells (Figure 5C,D, Table 1). This difference may be explained by the fact that chromatid cohesion emerging in the S-phase exerts an additional limiting effect on chromatin dynamics.

### 2.4. Cohesin Does Not Affect the Mobility of Repair Foci on the Time Scale Studied

One of the proposed functions of cohesin in DNA repair is to restrict the mobility of the ends of DSBs [18,21,25]. Thus, we decided to study the effect of cohesin depletion on the mobility of repaired DSBs. We used a genetically encoded fluorescent reporter—a fusion of the fluorescent protein FusionRed with a fragment of the repair factor 53BP1, called BP1-2 (amino acids 1220–1711 of 53BP1 [36])—to visualize DSBs. The repair factor 53BP1 is a component of the signaling cascade that occurs in response to DSBs and forms macroscopic foci in cells [51,52]. BP1-2 lacks most of the domains of 53BP1, but retains the oligomerization domain, as well as the Tudor domain and UDR motif required for binding chromatin during DSB repair [36,51,53]. We delivered the FusionRed-BP1-2 gene by lentiviral transduction into HCT116_RAD21_AID cells containing the CRISPR-Sirius imaging system.

We used the topoisomerase poison etoposide, which stabilizes the covalent complex of type II topoisomerases, preventing them from religating the break, to induce DSBs [54,55]. When exposed to etoposide (100 μg/mL), multiple fluorescent signals (foci) rapidly accumulated in the cells, the number of which reached a plateau by 15 min of incubation (Appendix A). Therefore, we incubated the cells with etoposide for 15 min to induce DSBs before microscopy, after which the medium was replaced with fresh medium. For RAD21 depletion, the cells were pre-incubated with auxins for 5 h. Next, we captured a time-lapse series of cells with visualized BP1-2 repair foci, localized individual foci in the images, obtained trajectories of geometric centers of foci, compensated trajectories for nuclear motility, and analyzed the mobility of the foci (Figure 6A, see also Appendix A). Microscopy was performed at the same time step (0.412 s) as in the case of microscopy of the C6 locus described in the previous sections. Only cells with at least five stably recognized trajectories were included in the analysis. The time-lapse duration was limited to 50 s (122 frames) because of the detected reporter’s photobleaching. For trajectory analysis, as in the case of the C6 locus, we calculated the mean displacement per frame, the gyration radius, and analyzed the parameters of the MSD curves. The median values of each parameter were first calculated for all tracks in a specific cell, and samples were compiled from such medians. Thus, the elements of the samples were not tracks, but cells, which allowed us to avoid biases because of the different numbers of recognized tracks in cells.

Cells expressing the FusionRed-BP1-2 reporter also contained the visualized C6 locus. Thus, we were able to discriminate cells in which the C6 locus had replicated (the S/G2 sample). Similarly to the previous C6 imaging experiments, cells were preincubated for 5 days on microscopy dishes to obtain a subset of cells in the G1 phase of the cell cycle. Using the same analysis of tracks as previously performed for the C6 locus, we found no significant changes in signal mobility upon RAD21 depletion, with no differences in either G1 or S/G2 cells (Figure 6B,C). This outcome suggests that cohesin does not limit the dynamics of repair foci at the time scales studied (subseconds to a minute). However, all measured parameters were smaller for BP1-2 foci than the previously determined parameters for the C6 locus (Table 2). This result indicates that repair foci are less mobile structures than point chromatin loci, which may be of functional importance in the context of DSB repair (see the Section 3).

## 3. Discussion

Although the functions of the cohesin complex in sister chromatid retention and loop extrusion are generally recognized, the role of cohesin in the spatial mobility of chromatin loci in human cells is yet unclear. Cohesin may reduce chromatin mobility by promoting chromatid cohesion and forming contact domains. On the other hand, contact domains are formed by constant nonequilibrium loop extrusion, which, on the scale of individual loops and topological domains, can promote local dynamics of chromatin loci. Thus, we obtained cells with an auxin-inducible degron system—in which auxin addition depletes the RAD21 protein, one of the main subunits of the cohesin complex—to test whether cohesin acts as a constraining factor of short-scale chromatin mobility or promotes chromatin dynamics. Using the CRISPR-Sirius system, we visualized a genomic locus (C6) in the obtained cells, which served as a model in this study.

We performed imaging at a speed of ~0.4 s/frame with a track length of ~1 min, which allowed us to study local chromatin dynamics in terms of the space covered by the visualized locus. This distinguishes our study from that of Mach et al. [15], in which the time-lapse imaging of TetO arrays in mouse embryonic stem cells was performed with a time step of 10 s and an imaging duration of 30 min. Using an unsynchronized cell population, we showed that cohesin depletion enhances the dynamics of the imaged locus, which was manifested by an increase in biophysical parameters such as the mean displacement, gyration radius, diffusion coefficient, and anomalous exponent. An increase in the diffusion coefficient upon RAD21 depletion was also reported by Mach et al. [15]. Interestingly, despite the difference in the experimental systems, the increase in the median value of the diffusion coefficient in the present work (approximately 1.5 times) was the same as reported by Mach et al. [15]. However, the data on the anomalous exponent differ: in our case, the anomalous exponent increased from ~0.2 to ~0.3, whereas the value of the anomalous exponent reported by Mach et al. was approximately 0.6 before and after RAD21 depletion [15]. This is most likely explained by the larger time scale on which Mach et al. performed the observation [15]. This difference indicates that cohesin has a more pronounced effect on the diffusion limitation of the chromatin locus at small space and time scales (our study). By contrast, at larger scales, this effect becomes less pronounced [15].

Although studies of unsynchronized cells provide valuable insights, they may mask cell cycle phase-dependent effects. Thus, we examined the effect of RAD21 depletion on the dynamics of the visualized locus in replicated and unreplicated chromatin separately. We found that cohesin acted as a constraint on the dynamics of the locus before and after its replication. This finding suggests that chromatid cohesion is not indispensable for the constraint on chromatin dynamics, and partial constraint is likely achieved by cohesin compacting chromatin into topologically associated domains through loop extrusion. Moreover, the effect of cohesin depletion on replicated chromatin was more pronounced than on non-replicated chromatin. In the latter case, the anomalous exponent did not change, and the other measured diffusion parameters showed less pronounced variations. The stronger effect of cohesin depletion on locus mobility in replicated chromatin can be explained by chromatid cohesion. However, other factors that differentially affect chromatin dynamics in different phases of the cell cycle cannot be ruled out. Perhaps, a more detailed differentiation of the effect of extrusion and cohesion on chromatin dynamics can be obtained using the SMC1^3D^ cohesin mutant, which has impaired cohesion activity but retains the extrusion ability [56]. Overall, our study shows that cohesin constrains the dynamics of the chromatin locus, even on small time and spatial scales. Nevertheless, we cannot rule out that a possible loop extrusion effect stimulating the spatial dynamics of the locus (e.g., when a loop is formed) occurs on smaller time scales or requires super-resolution microscopy to be detected.

The requirement of cohesin for DSB repair in post-replicative chromatin has been demonstrated in numerous studies on yeast and human cells [17,18,21,57,58,59,60]. Most of these studies were published before the extrusion activity of cohesin was discovered. Thus, cohesion and, consequently, restriction of the mobility of the break ends were considered possible functions of cohesin in repair [18,25,59,60,61]. This hypothesis is supported by the fact that knockdown of the SMC1 and RAD21 subunits of cohesin did not reduce the repair efficiency in the G1 phase of the cell cycle, but only in the G2 phase [18]. Furthermore, Gelot et al. found an increase in the probability of ligation of distant DSB ends, but not nearby DSB ends, upon knockdown of both cohesin (RAD21) and sororin, a cohesion stabilizing factor [21]. Sororin, like cohesin, is required for efficient DSB repair in the G2 phase, confirming the requirement of cohesion for efficient repair of post-replicative DSBs [61].

On the other hand, several possible cohesion-independent activities of cohesin in repair have been described in recent years. For example, Meisenberg et al. found a slight decrease in repair efficiency and an increased number of chromosomal rearrangements in the G1 phase of the cell cycle upon knockdown of the cohesin subunit SA2 [19]. It has also been shown that unidirectional loop extrusion on either side of a double-strand break promotes the propagation of a wave of H2AX phosphorylation, which is needed for the formation of the repair domain [26]. Additionally, cohesin, but not sororin, is required for full activation of Chk2 checkpoint kinase after DSB induction in human cells, not only in the S and G2 phases, but also in the G1 phase of the cell cycle [29].

We found that cohesin restricts chromatin dynamics in both post-replicative and unreplicated chromatin, suggesting that cohesin may stabilize chromatin loci during repair. Moreover, cohesion is not necessary for this process because the restrictive effect of cohesin is not only seen in S/G2 but also in G1 cells. The more pronounced effect of cohesin depletion in the S/G2 phases may explain why cohesin’s effect on repair efficiency was detected in human G1-phase cells in one study [19] but not in another [18].

In light of the possible role of cohesin in maintaining the spatial stability of DSBs, we also investigated the effect of cohesin depletion on the dynamics of repair foci visualized using the fluorescent reporter FusionRed-BP1-2. This reporter is a truncated version of the repair factor 53BP1, and like its full-length precursor, it forms microscopically distinguishable foci in the nucleus upon DSB induction [36]. We found no change in the mobility of FusionRed-BP1-2 foci upon cohesin depletion. The effect of cohesin depletion was absent in both unreplicated and replicated chromatin. It is important to note that our reporter allowed us to study the dynamics of fairly large repair foci rather than individual ends of DSBs. Our results indicate that repair foci are fairly stable structures, and their dynamics on the studied time scales are independent of cohesin. The spatial stability of repair foci can be explained by the fact that such structures, unlike point genomic loci, are macromolecular aggregates enriched in multiple intermolecular contacts within themselves [62,63]. In the case of 53BP1, contacts with chromatin are provided by this protein’s TUDOR domain and UDR motif, interacting with H4K20me2 and H2A(X)K15Ub, respectively (reviewed in [64,65]). In addition, 53BP1 can oligomerize [53,64,65]. Chromatin stabilization may be one of the functions of 53BP1-containing condensates formed on DSBs during repair, preventing undesirable recombination. Thus, a stabilizing effect may be maintained even in the background of cohesin depletion. However, we cannot conclude that 53BP1 exerts such a stabilizing effect in repair foci. In our case, 53BP1 (or more precisely, its derivative BP1-2) was only a way to visualize repair foci. Other proteins included in the visualized repair condensates may be responsible for spatial stability.

Our results at first glance contradict earlier data obtained in yeast cells by studying RAD52 repair foci at spontaneously occurring DSBs in the S-phase [66]. The mobility of such foci increased in the S-phase upon degradation of Scc1, a yeast homolog of RAD21. Similarly, breaks in the *MAT* locus in the S-phase also acquired greater mobility upon depletion of Scc1 [67]. This discrepancy may be explained by several factors. Previous live-cell imaging experiments have shown that DSBs in yeast cells are, in principle, more dynamic than DNA breaks in vertebrate cells [68,69,70,71,72]. In particular, the spatial stability of DSBs in human cells was demonstrated using the 53BP1-GFP reporter [71]. This difference can be explained by the fact that in actively dividing yeast cells, especially after replication, repair is predominantly achieved by homologous recombination, while in vertebrate cells, the dominant repair pathway is non-homologous end joining (NHEJ) [73,74]. The mobility of breaks in yeast cells may thus facilitate the search for a recombination partner [68,69]. At the same time, such mobility would be extremely dangerous in vertebrate cells, whose genome is rich in repetitive sequences, and would most likely lead to ectopic recombination and chromosomal aberrations. Interestingly, breaks repaired by homologous recombination in human cells in the G2 phase are more mobile and prone to clustering than breaks in the G1 phase, repaired by the NHEJ pathway [75]. It is worth noting that the reporter we used does not allow discrimination between repair pathways because 53BP1 binds both to breaks repaired via the NHEJ pathway and to breaks that will be further repaired via the homologous recombination pathway [52,76,77].

Although we did not detect an increase in the mobility of repair foci as integral structures, the dynamics of individual chromatin loci within such foci may increase with cohesin depletion. This issue could be studied in more detail by directing a nuclease (e.g., Cas9) to the C6 locus that we visualized. However, we have not managed to select a sufficiently effective guide RNA that would allow us to introduce and visualize breaks in our model locus.

We should note that our study is limited to a single model locus (C6), which, according to published PRO-Seq data [7], is expressed at a low level in HCT116 cells. Therefore, we cannot conclude that cohesin depletion has the same effect on the mobility of chromatin loci with different transcriptional states. It is worth discussing that the relationship between chromatin dynamics and transcriptional activity in mammalian cells is quite complex [78]. Targeted monitoring of individual loci has shown that transcriptional activation can both enhance [79] and restrict the dynamics of visualized genomic elements [80]. Methods aimed at studying the dynamics of the entire chromatin, such as single-nucleosome imaging [81,82,83,84] and dense optical flow reconstruction of chromatin mobility (Hi-D) [85], consistently indicate that peripheral heterochromatin exhibits significantly lower dynamics than chromatin in the nuclear interior. However, data on the relationship between chromatin mobility and its compaction within the nuclear interior remain contradictory. While Hi-D analysis did not reveal a correlation between chromatin packing density and motility [85], more recent work using single-nucleosome tracking suggests that a weak inverse correlation does exist [84]. Furthermore, analysis of single Cy3-dCTP-labeled nucleosome dynamics in early-replicating euchromatin suggests that nucleosomes in euchromatic regions are slightly more dynamic than in the nucleus as a whole [86]. Consistent with this, treatment with trichostatin A, a histone deacetylase inhibitor, increases nucleosome dynamics, particularly in denser chromatin regions [82,84]. It has also recently been shown that nucleosomes in H3K27ac-enriched chromatin indeed exhibit increased mobility [87].

At the same time, contrary to expectations, inhibition of transcription elongation via treatment with 5,6-Dichloro-1-β-D-ribofuranosylbenzimidazole (DRB) or α-amanitin, as well as depletion of RNA polymerase II using the auxin degron system, increases local nucleosome dynamics [82,83,84]. The dynamics-limiting effect of transcription can be explained by the formation of clusters or droplets of active RNA polymerases in the nucleus, which restrict the mobility of chromatin loci involved in these condensates [83,84]. Simultaneous labeling of active RNA polymerase II (RNAP2-Ser5ph) and imaging of single nucleosomes confirm the reduced mobility of nucleosomes associated with active RNA polymerase [87]. The opposing effects of H3K27 acetylation and active transcription on nucleosome dynamics suggest that chromatin poised for transcription (containing H3K27ac) and actively transcribed chromatin (containing RNAP2-Ser5ph) represent functionally distinct states with different dynamics [87]. Returning to our study and considering the findings described above, it would be valuable to investigate the effect of cohesin depletion on the dynamics of loci with varying transcriptional activity. Expanding the panel of CRISPR-Sirius–visualized chromatin loci with different transcriptional states could help address this question in future work.

## 4. Materials and Methods

### 4.1. Plasmid Construction

A guide RNA with the sequence GCAAGGTTCCATATTATATA was used for CRISPR/Cas9-mediated integration of the mAID degron sequence into the *RAD21* gene. The guide RNA sequence was taken from [88], with the first nucleotide replaced by G for higher transcription efficiency from the U6 promoter [89]. The guide RNA sequence in the form of a double-stranded DNA oligonucleotide with the appropriate overhangs was ligated into the pX330 vector [90] at the BstV2I restriction sites (the sequences of all primers and oligonucleotides used are given in Appendix A). The assembly of vectors for integration of the mAID degron was carried out in several stages. First, the mAID degron sequence and the linker sequence before it were amplified from the pMK289 plasmid (Addgene #72827, Watertown, MA, USA [39], kindly provided by N.R.Battulin, Institute of Cytology and Genetics SB RAS) with primers mAID_SalI_link_f + mAID_BstV2I_KpnI (see Appendix A), and cloned into the pUC18 plasmid [91] at the SalI and KpnI sites. Then, the T2A-Neo or T2A-Hygro sequences were amplified from the AAVS1-Neo-M2rtTA vector (Addgene #60843 [92]) and the ggh5_miniIAA7_Hygro vector (kindly provided by N.R.Battulin, Institute of Cytology and Genetics SB RAS) with the primers T2A_Neo_f_AsiGI + Neo_r_SacI and T2A_Hygro_f_AsiGI + Hygro_r_SacI, respectively, and cloned into the resulting vector at the BstV2I and SacI sites. Then, the homology arms were amplified from the HCT116 genomic DNA and inserted into the resulting vectors at the SalI and SacI sites. The left homology arm was amplified with HA_L_f + HA_L_r primers; the right homology arm was amplified using the HA_R_Neo_f or HA_R_Hygro_f primers, depending on the vector type, and the HA_R_r primer as a reverse primer in both cases. Vectors with homology arms were assembled by the Gibson assembly approach using the NEBuilder HiFi DNA Assembly kit (NEB, Ipswich, MA, USA). The resulting vectors were named pUC18_mAID_Neo_HA and pUC18_mAID_Hygro_HA. To assemble the lentiviral transfer vector with the OsTIR1 gene, the reading frame of MCP-HaloTag-NLS in the pHAGE-EFS-MCP-HALOnls vector (Addgene #121937 [30]) was replaced at the XbaI and MluI sites with the reading frame of OsTIR1 amplified from the pMK232 plasmid (Addgene #72834 [39], kindly provided by N.R. Battulin, Institute of Cytology and Genetics SB RAS) with the primers OsTIR_f_MluI + OsTIR_r_XbaI. The vector for delivering the FusionRed-BP1-2 gene was constructed on the basis of the Lenti_FusionRed vector available in our laboratory. The Lenti_FusionRed vector is a transfer vector for the assembly of second-generation lentiviral particles, which contains the FusionRed fluorescent protein gene under the control of the minimal EF-1α promoter. The BP1-2 sequence was amplified from the pcDNA5-FRT/TO-eGFP-53BP1 vector (Addgene #60813 [51], kindly provided by A.K.Velichko, Institute of Gene Biology of the Russian Academy of Sciences) with the BP1_Xho_f + BP1_Xba_r primers and cloned into the Lenti_FusionRed vector at the XbaI and XhoI sites. High-fidelity polymerase Q5 Hot start (NEB) was used for all preparative PCR. The sequences of all inserts in the vectors obtained during the work were confirmed by partial sequencing. Zip-archive with the annotated sequences of the pUC18_mAID_Neo_HA and pUC18_mAID_Hygro_HA vectors, as well as the transfer vectors for the lentiviral delivery of the OsTIR1 and FusionRed-BP1-2 genes, is provided in Appendix A. The lentiviral vectors described in our previous article [33] were used to deliver the genes for dCas9, MCP-sfGFP and two guide RNAs for visualization of the C6 locus (C6_sg1 and C6_sg2). The sequences of the recognition parts of the guide RNAs C6_sg1 and C6_sg2 for CRISPR-Sirius visualization were GTGAGTGCACAC and TGGGACACTATGATG, respectively [33].

### 4.2. Lentiviral Particles Production

HEK293T cells were cultured in DMEM medium with alanyl-glutamine (PanEco, Moscow, Russia) supplemented with 10% fetal bovine serum (FB-1001, Biosera, Cholet, France) and 1× antibiotic-antimycotic (L0010, Biowest, Bradenton, FL, USA) at 37 °C under 5% CO_2_. The day before transfection, 3 × 10^5^ cells were seeded into the wells of a six-well plate in 2 mL of medium. The next day, the cells were transfected with a mixture of plasmids pCMV-VSV-G (Addgene #8454 [93], kindly provided by A.P.Kovina, IGB RAS) and pCMV-dR8.2-dvpr (Addgene #8455 [93], kindly provided by A.P. Kovina, IGB RAS), encoding the proteins necessary for lentiviral assembly, as well as the required transfer vector with the gene of interest, 4 μg of each plasmid. The genes of interest were OsTIR1, dCas9, MCP-sfGFP, visualizing guide RNAs genes containing MS2 repeats, and the FusionRed-BP1-2 reporter gene. The transfection mixture was prepared using 400 μL of Opti-MEM I medium (Gibco, Waltham, MA, USA) with the addition of 10 μL of Turbofect reagent (R0531, Thermo Fisher Scientific, Waltham, MA, USA) according to the manufacturer’s protocol. After four days, the supernatants (suspensions of lentiviral particles) were collected from the cells, filtered through a 0.45 μm filter, and used for cell transduction or frozen at −80 °C until use.

### 4.3. Generation of HCT116 Cells with the Auxin-Inducible RAD21 Depletion and the CRISPR-Sirius System

HCT116 cells were cultured in DMEM medium with alanyl-glutamine (PanEco) supplemented with 10% fetal bovine serum (FB-1001, Biosera) and 1× antibiotic-antimycotic (L0010, Biowest) at 37 °C under 5% CO_2_. For the integration of degron into the genome of cells, the homology donor vectors pUC18_mAID_Neo_HA and pUC18_mAID_Hygro_HA, as well as the pX330 vector with guide RNA to the RAD21 gene were transfected on the Neon Transfection System (Thermo Fisher Scientific) using supplied kit reagents with 100 μL tips. The following electroporation protocol was used: voltage 1100 V, pulse duration 30 ms, two pulses. For transfection, 4 μg of each plasmid and 1 million HCT116 cells were used. Transfected cells were selected on a medium with geneticin G-418 (Capricorn Scientific, Ebsdorfergrund, Germany) at a concentration of 800 μg/mL. Transfected cells were cultivated on a medium with G-418 until complete death of control non-transfected cells, cultivated in parallel. Then, G-418 resistant cells were subcultured onto a medium with hygromycin B (Invitrogen, Waltham, MA, USA) at a concentration of 150 μg/mL. Transfected cells were cultivated on a medium with hygromycin B until complete death of control non-transfected cells, cultivated in parallel. After selection, the cells were cloned. PCR analysis of the integration of the mAID degron into the RAD21 gene in the obtained cell clones was carried out in two steps. First, PCR was performed with the RAD21_f and RAD21_r primers flanking the integration point (sequences of all primers are provided in Appendix A). PCR was performed with DreamTaq polymerase (Thermo Fisher Scientific) using the direct PCR method [94]. For this, 10–50 thousand cells were first lysed in 10 μL of 2× buffer from the DreamTaq polymerase kit with the addition of dNTPs at a concentration of 0.4 mM each and proteinase K (EO0491, Thermo Fisher Scientific) at a concentration of 0.4 mg/mL. The samples were incubated at 60 °C for 20 min, and then the proteinase was inactivated by heating at 95 °C for 5 min. After that, the samples were cooled, and 10 μL of a solution containing 0.5 μM of each primer and 0.8 U of DreamTaq polymerase were added to each tube. PCR was carried out according to the program: 95 °C 5 min—(95 °C 20 s—58 °C 20 s—72 °C 3 min) × 40—72 °C 5 min. Clones in which the PCR product of the integrated construct was detected and the PCR product of the wild type RAD21 allele was not detected were further validated using direct PCR with primers for different regions of the integrated constructs: the degron sequence (primers RAD21_f + mAID_r), and the sequences of antibiotic resistance open reading frames (primers Neo_f + RAD21_r, and Hygro_f + RAD21_r). In the cell clone selected for further work, the sequence of the end of the RAD21 reading frame and the following mAID degron was confirmed by Sanger sequencing of PCR products amplified from the genomic DNA of this clone with primers RAD21_f + mAID_r and RAD21_f2 + T2A_r using Tersus high-fidelity polymerase (Evrogen, Moscow, Russia). Then, the OsTIR1 gene was integrated into the genome of the selected cell clone by lentiviral transduction. For this, 4 × 10^5^ cells were mixed with 100 μL of a suspension of the corresponding lentiviral particles in 2 mL of DMEM medium with the addition of the Polybrene reagent (TR-1003, Sigma-Aldrich) at a concentration of 8 μg/mL. Four days later, the cells were sorted one per well in a 96-well plate using a FACSAria SORP cell sorter (BD Biosciences, Franklin Lakes, NJ, USA). For clone screening, genomic DNA was isolated from each clone using the DU-250 reagent kit (Biolabmix, Novosibirsk, Russia). PCR was performed with the isolated genomic DNA with the OsTIR_f1 + OsTIR_r1 primers using HS Taq polymerase (Evrogen) with the following program: 95 °C 5 min—(95 °C 15 s—58 °C 20 s—72 °C 30 s) × 32—72 °C 5 min. One of several cell clones with confirmed integration of the OsTIR1 gene was named HCT116_RAD21_AID and was used in all subsequent work. Integration of the genes encoding the CRISPR-Sirius system components into HCT116_RAD21_AID cells (dCas9, MCP-sfGFP, and guide RNA genes) was performed by lentiviral transduction as described previously [33]. The locus designated as C6 with coordinates chr6: 157,310,367–157,314,361 (hg38) was used as the target. In the case of integration of the Sirius guide RNA genes into HCT116_RAD21_AID cells, selection on hygromycin B was not performed, since resistance to this antibiotic was already used during the integration of the mAID degron. Generation of control HCT116 cells with the CRISPR-Sirius imaging system but without the AID system was also described previously [33].

### 4.4. Integration of the FusionRed-BP1-2 Gene

Integration of the FusionRed-BP1-2 double-strand break reporter gene into the genome was accomplished by lentiviral transduction. For this purpose, 4 × 10^5^ cells were mixed with 100 μL of the corresponding lentiviral particle suspension in 2 mL of DMEM medium supplemented with Polybrene reagent (TR-1003, Sigma-Aldrich, St. Louis, MO, USA) at a concentration of 8 μg/mL. Four days after transduction FusionRed-bright cells were sorted on a FACSAria SORP cell sorter (BD Biosciences) using the 85 um nozzle and the corresponding pressure parameters. The FusionRed fluorescence was detected using 561 nm laser excitation and a set of 595LP + 610/20BP emission filters. Non-transduced HCT116 cells were used as a negative control to set the gate for sorting.

### 4.5. Western Blotting

Cells were seeded into the wells of a six-well plate at 1.5 million cells in 2 mL of medium the day before the experiment. The next day, auxins were added to the cells. A mixture of sodium salts of indoleacetic acid (IAA sodium salt, I5148, Sigma-Aldrich, dissolved in water to 500 mM stock) and α-naphthylacetic acid (α-NAA, N0640, Sigma-Aldrich, dissolved in 1 M NaOH and diluted with water to 500 mM stock) at a concentration of 500 μM each were used as auxins in all experiments in this work. After the indicated time, the cells were removed with a trypsin-EDTA solution (PanEco), washed with DPBS with the addition of auxins at a working concentration (DPBS without auxins was used for untreated cells). The cells were lysed in 100 μL RIPA buffer (50 mM Tris-HCl pH 7.4, 150 mM NaCl, Nonidet P-40 1%, SDS 0.1%, sodium deoxycholate 0.5%) with the addition of 5 μL Protease Inhibitor Cocktail (P8340, Sigma-Aldrich) and PMSF to 1 mM. The cells were lysed for 1 h in a cold room with constant rotation at 30 rpm. Then the lysate was centrifuged at 16,000× *g* for 20 min at +4 °C. Supernatants were mixed with 5× sample loading buffer (160 mM Tris-HCl pH 6.8, 5% SDS, 25% glycerol, 50 mM DTT, 0.05% bromophenol blue), and 12 μL of samples were loaded into wells of a 1 mm gel (6% stacking gel–8% separating gel) and separated by standard SDS-PAGE. Prestained molecular weight marker RAV-11 (Biolabmix) was used to monitor protein movement. The proteins were then transferred to a Hybond-C Extra nitrocellulose membrane (Amersham Biosciences, Amersham, UK) by wet transfer in Towbin buffer (25 mM Tris, 192 mM glycine, 10% methanol) for four hours at a constant voltage of 80 V using a Mini Trans-Blot Cell (Bio-Rad, Hercules, CA, USA). After transfer, the membrane was stained with Ponceau S (0.2% solution in 5% acetic acid) for subsequent normalization to total protein. The stained membrane was photographed using a ChemiDoc Touch Imaging System (Bio-Rad). Ponceau S was washed with TBS buffer (20 mM Tris-HCl pH 7.6, 150 mM NaCl) twice for 5 min, after which the membrane was blocked for one hour at room temperature (RT) in TBS buffer with 0.05% Tween-20 (TBST) with 5% skim milk. The membrane was washed with TBST buffer once for 5 min and then incubated overnight at +4 °C in a solution of primary antibodies against RAD21 (ab992, Abcam, Cambridge, UK) diluted 1:4000 in TBST with 1% bovine serum albumin (BSA). The next day, the membrane was washed four times for 5 min with TBST buffer, after which it was incubated in a solution of HRP-conjugated secondary antibodies (ab205718, Abcam) at a dilution of 1:5000 in TBST buffer with 1% BSA for two hours at room temperature. The membrane was washed three times for 5 min with TBST buffer and once more with TBS buffer, after which it was developed using the Clarity Western ECL Substrate reagent kit (Bio-Rad) according to the manufacturer’s instructions. The membrane was photographed on a ChemiDoc Touch Imaging System (Bio-Rad) at an exposure that did not lead to the appearance of overexposed pixels. The unprocessed (raw) image was saved in .tif format. The ECL image was then analyzed in the FiJi software [95] (version 2.16.0/1.54p). First, the background intensity value was measured for the empty lane and several membrane areas free of protein bands. Then, the average background mean gray value was subtracted from all image pixels using the ‘Subtract’ tool. Then, the target band was outlined, after which the integrated intensity value in the selected area was measured. The image of the membrane stained with Ponceau S was processed similarly. However, in this case, not a separate band was selected, but the entire lane (except for the empty areas at the top and bottom of the lane). To calculate the RAD21 protein content in the sample normalized to the total protein (in linear scale arbitrary units), the signal intensity value of the target protein was divided by the staining intensity value of the total protein in the lane. Image contrast was not adjusted at any stage of image analysis.

### 4.6. Analysis of Cell Cycle, Mitotic Index and Growth Curves

To analyze cell cycle phase distribution, cells were detached with trypsin-ETDA solution, washed in DPBS and fixed with cold (−20 °C) 70% ethanol overnight at +4 °C. The next day, fixed cells were washed with DPBS and then stained with propidium iodide solution at a concentration of 30 μg/mL in DPBS or Hoechst33342 solution at a concentration of 5 μg/mL in DPBS. In both cases, RNase A (Thermo Fisher Scientific) was added to the staining solution to a concentration of 5 μg/mL. Staining was performed overnight at +4 °C. Stained cells were analyzed using a FACSAria SORP instrument (BD Biosciences). PI was detected using 488 nm laser excitation and a set of 550LP+582/15BP emission filters. Hoechst33342 was detected using 407 nm laser excitation and a 450/50BP emission filter. For the analysis, cells were initially gated from debris based on forward and side scatter (FSC-A vs. SSC-A). Additionally, a gate was set in the coordinates of the width and height of the FSC peak (FSC-W vs. FSC-H) to select single cells. For the gated events a histogram of the PI or Hoechst33342 fluorescence intensity distribution on a linear scale was plotted.

To determine the proportion of mitotic cells (i.e., the mitotic index) and the proportion of cells with abnormal nuclei, 1.5 × 10^5^ cells in 400 μL of medium were seeded in the center of a 35 mm glass-bottom dish (on the glass) the day before the experiment. The next day, a warm DMEM medium was added to the cells to 1.5 mL. Auxins were also added to indicated samples for 12 or 24 h. As in other experiments, a mixture of IAA and α-NAA, at a concentration of 500 μM each, was used as auxins. Before microscopy, 5 μL of Hoechst33342 Ready Flow reagent (Invitrogen) were added to the cells. After 10 min, the cells were analyzed on an Olympus Fluoview FV3000 confocal microscope (Olympus, Tokyo, Japan) equipped with a Olympus 60× UplanXApo oil immersion objective (numerical aperture (NA) = 1.42) and Olympus FV31-HSD detector. The Hoechst33342 fluorescence was excited using a Coherent OBIS405 laser. Microscopy was performed at 37 °C in a humidified atmosphere with 5% CO_2_ in a stage-top incubator with an STX Temp and Flow module (Tokai Hit, Fujinomiya, Shizuoka, Japan). Z-stacks of several random fields of view were taken for each sample.

To analyze growth curves, 3 × 10^4^ cells in 500 μL of DMEM medium were seeded into wells of a 24-well plate. The next day, IAA and α-NAA were added to a portion of the wells of each culture (HCT116 or HCT116_RAD21_AID) at a concentration of 500 μM each. Then, over several days, one well of cells per day was detached with a trypsin-EDTA solution and the number of cells in the well was determined using a hemocytometer. The resulting cell numbers were normalized on the number of cells on the day of auxin addition.

### 4.7. ChIP-Seq

Cells (2 × 10^6^) were seeded in 2 mL of DMEM medium into the wells of a six-well plate one day before the experiment. The next day, IAA and alpha-NAA (500 μM each) were added to cells for 6 h, after which the medium was removed and the cells were fixed in 1 mL of 1% formaldehyde solution in serum-free DMEM for 10 min. To neutralize formaldehyde, glycine was added to a concentration of 125 mM for 2 min. Then the cells were detached with a cell scraper. The cells were washed with DPBS and then lysed in 500 μL of cold RIPA buffer (50 mM Tris-HCl pH 8.0, 150 mM NaCl, 2 mM EDTA, 0.5% SDC (sodium deoxycholate), 0.1% SDS, 1% NP-40) with the addition of PMSF to 1 mM and 5 μL of Protease Inhibitor Cocktail (P8340, Sigma-Aldrich). Lysates were sonicated on a Branson Digital Sonifier 450 (Branson Ultrasonics Corporation, Danbury, CT, USA) in four series of 10 pulses in the mode of 7 s ON/30 s OFF at a power of 10%. Breaks between series were 4 min. The cells were kept on ice, pre-cooled to −20 °C, during the entire sonication period. Then the cell debris was sedimented by centrifugation for 5 min at 20,000× *g* at 4 °C. Supernatants (chromatin fraction) were transferred to Amicon Ultra 0.5 mL 30 K filter units (Merck Millipore, Burlington, MA, USA) and centrifuged for 5 min at 14,000× *g* at 4 °C. The solution remaining in the filter was transferred to a new tube and its volume was brought to 1 mL with IP buffer (15 mM Tris-HCl 8.0, 0.01% SDS, 1% Triton X-100, 2 mM EDTA, 150 mM NaCl, 1 mM PMSF, Protease Inhibitor Cocktail (P8340, Sigma-Aldrich) 10 μL per 1 mL buffer volume). 100 μL were taken as Input and frozen at −20 °C. The volume of samples was adjusted to 1 mL with IP buffer, and 2 μL of anti-RAD21 antibodies (ab992, Abcam) were added to the samples for overnight incubation under constant rotation at +4 °C. The next day, 25 μL of Protein A/G ChIP grade magnetic beads (Thermo Fisher Scientific) were added to the samples, after being pre-washed in 1 mL of RIPA buffer with 0.5% BSA overnight under constant rotation at +4 °C. The samples were bound to the beads for 6 h under constant rotation at +4 °C. The beads were then washed sequentially in WBA (20 mM Tris-HCl 8.0, 0.1% SDS, 1% Triton X-100, 2 mM EDTA, 150 mM NaCl), WBB (same as WBA, except 500 mM NaCl was used), and WBC (10 mM Tris-HCl 8.0, 1% SDC, 1% NP-40, 1 mM EDTA, 250 mM LiCl) buffers for 10 min under constant rotation at +4 °C. Formaldehyde cross-links and proteins were then destroyed by incubating the beads in TE buffer (10 mM Tris-HCl pH 8.0, 1 mM EDTA) supplemented with proteinase K to 0.5 μg/μL and SDS to 0.5% overnight at 65 °C on a shaker at 1400 rpm. Input was decrosslinked in the same way. DNA was further purified from samples by phenol-chloroform extraction, precipitated with ethanol and sodium acetate with the addition of yeast tRNA and glycogen as co-precipitants. The DNA pellets were dissolved in 50 μL of 10 mM Tris-HCl buffer pH 8.0. RNase A (R1253, Thermo Fisher Scientific) was added to the samples to a concentration of 0.2 mg/mL for 30 min at 37 °C. DNA was then further purified using MagPure A4 XP Beads (Magen, Guangzhou, Guangdong, China) and eluted in 50 μL of 10 mM Tris-HCl buffer pH 8.0. DNA ends were then repaired in a 100 μL reactions containing 50 μL sample, 0.5 mM each dNTP, 5 μL T4 polynucleotide kinase (M0201L, NEB), 4 μL T4 DNA polymerase (M0203L, NEB), 1 μL Klenow fragment (M0210L, NEB), in 1× T4 DNA ligase buffer (Thermo Fisher Scientific). Reactions were incubated for 30 min at RT, after which DNA was purified from the reactions using MagPure A4 XP Beads (Magen) and eluted in 50 μL 10 mM Tris-HCl pH 8.0. 100 ng of Input DNA was treated similarly. Samples were then A-tailed. For this, 100 μL reactions containing 50 μL of sample, 0.5 mM dATP, and 5 μL of exo(−) Klenow fragment (M0212S, NEB) in 1× NEBuffer 2 (NEB) were mixed. The reactions were incubated for 30 min at 37 °C, after which DNA was purified from the reactions using MagPure A4 XP Beads (Magen) and eluted in 30 μL of 10 mM Tris-HCl pH 8.0. Adapters were then ligated to the samples in 50 μL reactions containing 30 μL of sample, 0.5 μL of Illumina TrueSeq adapters, and 10U of T4 DNA ligase (EL0011, Thermo Fisher Scientific) in 1× buffer supplied with T4 DNA ligase. Reactions were incubated overnight at room temperature, after which DNA was purified from the reactions using MagPure A4 XP Beads (Magen) and eluted in 30 µL of 10 mM Tris-HCl pH 8.0. Libraries were then amplified to achieve sufficient DNA for sequencing. Six 25 µL reactions were performed with each sample. Each reaction contained 4 µL of the appropriate sample DNA, 0.2 mM of each dNTP, Illumina PE1.0 and PE2.0 primers to 0.5 µM, and 0.5 U Q5 HotStart High-Fidelity DNA Polymerase (NEB) in the enzyme buffer supplied. Amplification was performed using the following program: 98 °C for 30 s, then the required number of cycles (98 °C 10 s—67 °C 15 s—72 °C 30 s), and then a final extension by 1 min at 72 °C. The number of PCR cycles was 9 for Input and 14 cycles for samples after immunoprecipitation. The required number of cycles was determined in a preliminary PCR carried out under the same conditions, but with sampling after certain cycles and monitoring the degree of DNA amplification by electrophoresis in agarose gel. After preparative amplification, reactions with one type of template were combined, and DNA from the reactions was purified using MagPure A4 XP Beads (Magen) and eluted in 40 μL of 10 mM Tris-HCl pH 8.0. Quality control of the samples was performed by electrophoresis in agarose gel. In case of detection of adapter dimer bands, the DNA libraries were additionally purified from the gel (the smear region between the 200–1000 bp markers) using the Cleanup Mini kit (Evrogen). DNA was then further purified using MagPure A4 XP Beads (Magen) and eluted in 30 µL 10 mM Tris-HCl pH 8.0. The resulting libraries were sequenced on an Illumina Novaseq 6000 instrument (San Diego, CA, USA) to generate 100 bp paired-end reads.

Sequencing data were processed on the Galaxy portal (https://usegalaxy.eu/ [96], accessed on 11 April 2025). During processing, reads were first trimmed using Trimmomatic (version 0.39 [97]). Illumina adapters clipping was performed using built-in TrueSeq3-PE adapter sequences with the following parameters: seed mismatches = 2, palindrome clip threshold = 30, simple clip threshold = 10, minAdapterLength = 4, keepBothReads = True. Leading low quality or N bases were removed (below quality 3). Sliding window quality trimming was performed with averaging across 4 bases and average required quality of 15. Finally, only reads with a minimum length of 20 bases were preserved. The reads remaining paired after trimming were aligned to the human reference genome, assembly GCA_000001405.15_GRCh38_no_alt_analysis_set, downloaded from NCBI (https://ftp.ncbi.nlm.nih.gov/genomes/all/GCA/000/001/405/GCA_000001405.15_GRCh38/seqs_for_alignment_pipelines.ucsc_ids/, accessed on 13 April 2025). Bowtie2 (version 2.5.3 [98]) was used for alignment. The alignment was performed with the --maxins = 1000 parameter, as well as with the --no-mixed, --no-discordant, and --very-sensitive (end-to-end) parameters active. The other parameters had their default values. Duplicate reads were then removed with MarkDuplicates tool (version 3.1.1.0) from Picard toolset with default parameters. To construct coverage profiles normalized to Input, the bamCompare tool from deepTools (version 3.5.4 [99]) was used. Profiles were built with bin size = 10 bp. Minimum mapping quality for reads was set to 5. Only the first read in each read pair was used to plot profiles (SAM flag of 64). Paired reads were extended to the corresponding fragment lengths with a paired-end extension option (--extendReads). SkipNAs option was used to skip non-covered regions. Each sample was normalized to fragments per kilobase per million mapped reads (RPKM). Normalization on Input control was carried out by calculating the ratio of the number of reads. Profiles in bigwig format were visualized using the IGV software without windowing option (version 2.19.2 [100]). MACS2 (version 2.2.9.1 [101]) with Input control was used to search for RAD21 binding peaks in HCT116 cells. The value of effective genome size was 2,805,636,231. Peak definition was based on q-value with a minimum of 0.05. Peak-averaged RAD21 binding profiles and heat maps were constructed using computeMatrix and plotHeatmap tools from deepTools toolset (version 3.5.4). The Input control-normalized RAD21 binding profiles in the corresponding samples obtained using the bamCompare tool (as described above) were used as profiles. Averaging was performed over the coordinates of peak summits obtained using MACS2 (as described above). ComputeMatrix tool was launched in reference-point mode with the following parameters: --binSize = 10, --beforeRegionStartLength = 1000, --beforeRegionStartLength = 1000, --missingDataAsZero and --skipZeros and default values of other parameters. RAD21 binding peaks were sorted in the shown heatmap according to the mean binding intensity in HCT116 cells while plotting with the plotHeatmap tool. The summary plots above the heatmaps for each sample represent the mean RAD21 binding intensity across all peaks found in HCT116 cells.

### 4.8. PRO-Seq Data Analysis

PRO-Seq data from [7] for HCT116 cells containing an auxin degron system were obtained from the GEO database, accession number GSE106886. Specifically, the file GSE106886_Rao-2017-Genes.rpkm.txt, containing RPKM values, was used to compare the expression levels of the *TMEM242* and *GAPDH* genes. For this analysis, the RPKM values for these genes were converted to TPM values and averaged across samples of untreated auxin degron cells. To compare *TMEM242* gene expression between auxin-treated and untreated cells, the file GSE106886_Rao-2017-RAD21notreat_vs_RAD21treat.Genes.DESeq2.txt, which contains DESeq2 differential gene expression analysis results, was used.

### 4.9. C6 Locus Visualization and Tracking

The day before microscopy, 5 × 10^5^ cells in 400 μL of medium were seeded in the center of a 35 mm glass-bottom dish (on the glass). The next day, a warm DMEM medium was added to the cells to 1.5 mL. Auxins were also added to indicated samples 5 h before microscopy. In the case of long-term cultivation (to accumulate cells in the G1 phase of the cell cycle), the cells were also seeded on a glass dish, but in the amount of 4 × 10^5^ cells per dish in 400 μL of medium. The next day, the cells were supplemented with medium to 1.5 mL and the cells were incubated for another four days. After 4 days, the cells were treated with auxins (if necessary) 5 h before microscopy. Next, a time-lapse series of images in one confocal layer was recorded using an Olympus Fluoview FV3000 microscope (Tokyo, Japan) equipped with a 60× UplanXApo oil immersion objective (NA = 1.42) and FV31-HSD detector. Imaging was performed with a time step of 0.412 s, a duration of 189 frames, and a resolution of 256 × 256 pixels. The pinhole size was set to 2 AU. To increase spatial resolution, the ROI size was set so that it included only one or two cells (resolution 103 nm/pixel). During microscopy, the cells were kept at 37 °C in a humidified atmosphere with 5% CO_2_ in a stage-top incubator with an STX Temp and Flow module (Tokai Hit). The total microscopy time for each dish did not exceed 90 min. sfGFP fluorescence was excited with a Coherent OBIS488 laser (Coherent, Santa Clara, CA, USA). The obtained time-lapse series were processed in the Fiji program [95]. First, Bleach correction was performed in the Histogram matching mode. Signals were registered using the TrackMate plugin [102,103] (version 7.13.0) using a DoG detector with sub-pixel localization. A signal radius of 0.6 μm was used, less often—0.5 μm for closely located replicated signals. The quality threshold was set for each image individually for optimal signal recognition depending on the signal brightness and signal-to-noise ratio. Tracking was performed using a simple LAP tracker with the parameters Linking max distance = 0.5 microns, Gap-closing max distance = 0.5 microns, Gap-closing max frame gap = 5. In case of tracking errors (e.g., if the trajectories of a replicated pair of signals were mixed, or if one track was divided into two), manual correction of the tracks was performed. Tracks of at least 180 frames in length were taken into further analysis. Missing points (no more than 5 for valid tracks) were completed by interpolation. To obtain nuclear tracks, a Thresholding detector and a simple LAP tracker were used. The intensity threshold was set individually for each cell, since the nuclei had different fluorescence intensities. During the analysis, the signal coordinates were initially converted to the coordinate system of the nucleus center to compensate for the movement of the nucleus. Next, for each track, a set of *MSD*(*t*) values was calculated using the following formula [31,32]:$$MSD\left(k∆t\right)=\frac{1}{\left(n-k+1\right)}\sum_{m=0}^{n-k}{\left|p\left(m∆t+k∆t\right)-p\left(m∆t\right)\right|}^{2},$$
where ***p***(*t*) is the position vector of the signal in XY coordinates of the center of the nucleus at time *t*, d*t* is the time interval between frames, *n* is the number of the last frame in the track (zero-based), *k* varies from 1 to n (the product *k*Δ*t* is designated as time lag). The first 20% of the points of the MSD curve of each track were approximated by the following equation [31,32]:$$MSD(t)=4D_{app}\times t^{\alpha }$$

From this equation, the values of the generalized diffusion coefficient *D_app_* and the anomalous exponent were obtained. Since the dimension of *D_app_* in such an equation is μm^2^/s*^α^*, i.e., depends on the value of *α* and varies among different tracks, the value of the diffusion coefficient was transformed using the equation *D*(*t*) = *D_app_***t^α^*^−1^ [104] for comparison between different tracks. At *t* = 1 s, *D* (1 s) is numerically equal to *D_app_* and has the dimension μm^2^/s. This value of the diffusion coefficient can be interpreted as the value of the diffusion coefficient on a time scale of 1 s. These are the values of the diffusion coefficient given in the article.

The mean displacement per frame was calculated using the following formula:$$Mean\;displacement=\;\frac{1}{n}\sum_{m=0}^{n-1}\left|p\left(m∆t+∆t\right)-p\left(m∆t\right)\right|$$

The notations for *n* and ***p***(*t*) here are the same as in the formula for calculating MSD (above). The gyration radius of a track was calculated using the formula [31]:$$R_{g}=\sqrt{\frac{1}{\left(n+1\right)}\sum_{m=0}^{n}{\left|p\left(m∆t\right)-p_{c}\right|}^{2}},$$
where ***p_c_*** is the position vector of the geometric center of the trajectory points (the notations of *n* and ***p***(*t*) are the same as in the formula for calculating MSD above). The position of the trajectory center was calculated using the formula:$$p_{c}=\frac{1}{\left(n+1\right)}\sum_{m=0}^{n}p\left(m∆t\right),$$
where *n* is the number of the last frame in the trajectory (zero-based).

### 4.10. FusionRed-BP1-2 Foci Visualization and Tracking

Cells were seeded on 35 mm glass-bottom dishes in the same way as described in the previous section for visualization of the C6 locus. To induce double-strand breaks, etoposide (E1383, Sigma) was added to the cells on the microcopy dish at a concentration of 100 μg/mL 15 min before microscopy. Then the cells on the dish were washed twice with 1 mL of warm DMEM medium, after which 1 mL of a new warm complete medium was added to the cells, in which the cells were further examined. In the case of cells pretreated with auxins, a medium supplemented with auxins at the working concentration was used for washing and subsequent microscopy. Visualization of FusionRed-BP1-2 foci for recording their tracks was carried out with the same system parameters as in the case of the C6 genomic locus, except for using a different laser—Coherent OBIS561, as well as with a shorter time-lapse duration—122 frames (the time step was the same—0.412 s). After recording the track, the state (replicated or not replicated) of the C6 loci in the given cell was also mentioned using the same parameters as in the case of visualization of the C6 locus for recording tracks (but in this case, images were not recorded). The total microscopy time for each dish did not exceed 1 h from the moment of adding etoposide. The resulting time-lapse series were processed in the Fiji software [95] (version 2.16.0/1.54p). Processing of the images began with Bleach correction in the Histogram matching mode. Then, the part of the image outside the cell nucleus was filled with background color to process only the signals in the nucleus of the selected cell. To improve signal recognition, the images were processed with the difference in Gaussians approach. To do this, the Gaussian Blur filter with sigma (radius) parameters of 2 and 4, respectively, was applied to two duplicates of the original time-lapse, after which the series of images processed with radius 4 was subtracted from the series of images processed with radius 2, using the Image Calculator tool in ’Subtract’ mode. Then the image was converted to binary using the Auto Threshold tool (version 1.18.0) with the following parameters: Method—‘Default’, ‘ignore black’ option enabled, ‘white objects on black background’ option enabled. Signal recognition was then performed using the TrackMate plugin using Mask detector on binarized time-lapses with the following parameters: ‘simplify contours’: false, minimum quality—3.9. Tracking of the recognized signals was performed using the Simple LAP tracker with the following parameters: Linking max distance = 0.2 microns, Gap-closing max frame gap = 3, Gap-closing max distance = 0.4 microns, the minimum number of trajectory points is 110. Only tracks with a total gap size of no more than 5 frames were taken into further analysis (missing points were completed by interpolation). For tracking nuclei, a threshold detector and the Simple LAP tracker of the TrackMate plugin were used. The intensity threshold was set individually for each cell, since the nuclei had different fluorescence intensities. The signal tracks were converted to the coordinate system of the nucleus center to compensate for the nucleus motion. Further analysis of the tracks was carried out as in the case of the C6 signal tracks, with the only difference that first, for each nucleus with at least five valid trajectories, the median values of all track parameters (*D*, *α*, mean displacement, gyration radius) in this nucleus were calculated. These median values were compared across different experimental conditions. Also in this case, the first 30% of the MSD curve points were used to calculate *D_app_* and *α* to ensure the same time lags coverage as in the case of the analysis of the C6 locus dynamics.

To analyze the kinetics of etoposide-induced double-strand break accumulation, 4 × 10^5^ cells in 400 μL of medium were seeded in the center of a 35 mm glass-bottom dish (on the glass). The next day, the cells were supplemented with medium to 1.5 mL. The cells were then examined on an Olympus Fluoview FV3000 microscope equipped with a 60× UplanXApo oil immersion objective (NA = 1.42) and FV31-HSD detector. During microscopy, the cells were kept at 37 °C in a humidified atmosphere with 5% CO2 in a stage-top incubator with an STX Temp and Flow module (Tokai Hit). After setting the focus, etoposide was added to the cells at a concentration of 100 μg/mL, and time-lapse was immediately started (the first frame was taken 3 min after etoposide was added), with a time interval of 3 min. The imaging was carried out at a resolution of 1024 × 1024 pixels, 174 nm/pixel. At each time point, several Z-steps were recorded with a step size of 1 μm. The images were processed in the Fiji software (version 2.16.0/1.54p). First, a maximum intensity Z-projections were made in the FusionRed channel. The number of signals was determined in each cell individually using the Find Maxima tool with the ‘Strict’ parameter. The ‘Prominence’ parameter was set individually for each cell for optimal signal recognition.

## 5. Conclusions

Using the auxin-inducible degron system and CRISPR-Sirius imaging technology, we demonstrated that cohesin acts as a restricting factor on local chromatin locus dynamics in human cells. Cohesin’s constraining of chromatin dynamics occurred in both replicated and unreplicated chromatin, indicating that cohesion is not required per se. However, cohesion acts as an additional constraint on the dynamics in replicated chromatin. In contrast to the point chromatin locus, repair foci are more stable structures, and their dynamics on time scales < 1 min are independent of cohesin activity.

## Figures and Tables

**Figure 1 ijms-26-08837-f001:**
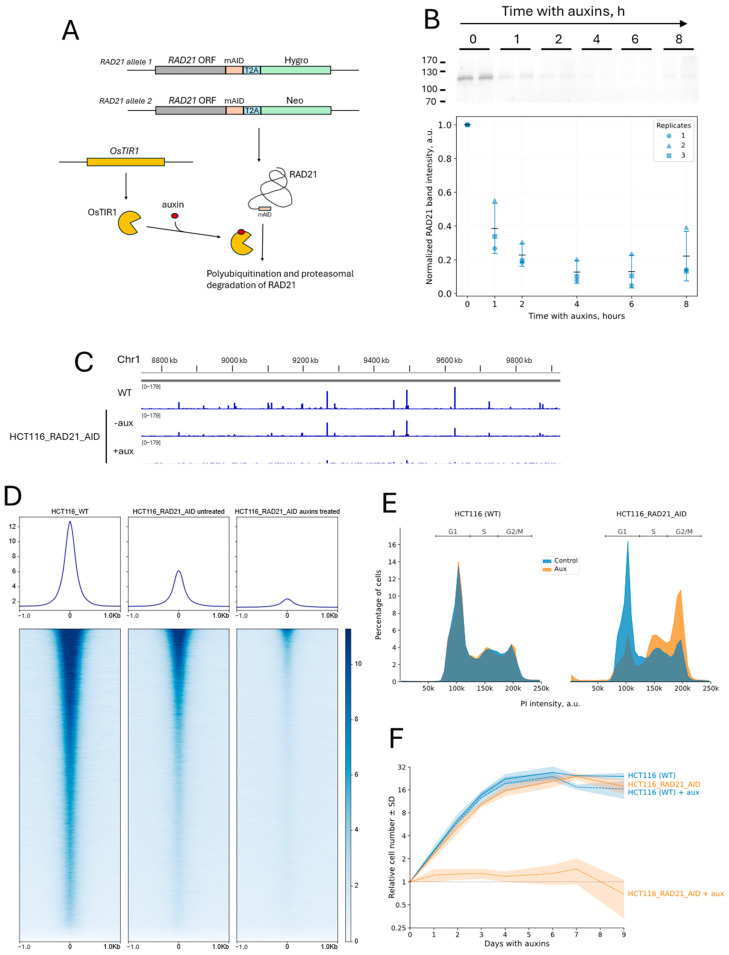
Cell line with auxin-inducible depletion of the RAD21 cohesin subunit. (**A**) Schematic of the AID system for RAD21 depletion. The mAID degron sequence was added to the end of the RAD21 reading frame in the genome of HCT116 cells. Antibiotic-resistance-reading frames (Neo and Hygro) allowed the selection of cells with biallelic integration of the degron. The cells also expressed the ubiquitin ligase OsTIR1, which targets RAD21 for proteasomal degradation upon addition of auxins. (**B**) Western blot analysis of the kinetics of RAD21 depletion upon addition of IAA and α-NAA (500 μM each, standard concentration in all experiments in this work). A fragment of the membrane with the RAD21 band is shown at the top. Each time point was applied in duplicate. The positions of the protein marker bands (in kDa) are marked; the arrow marks the position of the target RAD21 band. The graph below shows the mean RAD21 signal normalized to the total protein content in the lane at the indicated time points. For each replicate, intensity values were normalized to the 0 h value. Black horizontal bars represent the mean of three biological replicates at each time point; error bars denote ± standard deviation. Loading controls and full membrane images for all replicates are provided in Appendix A. (**C**) Input-normalized RAD21-binding profiles (ChIP-Seq) in wild-type (WT) HCT116 cells and HCT116_RAD21_AID cells in the absence of auxins (−aux) and after incubation with auxins for 6 h (+aux). The profile is shown for a region of chromosome 1 as an example. The coordinates correspond to the hg38 genome assembly. (**D**) Averaged RAD21 binding profiles in HCT116 (WT) cells, HCT116_RAD21_AID cells in the absence of auxins, and after 6 h of incubation with auxins. The profiles were averaged over all RAD21 peaks detected in HCT116 (WT) cells. Heat maps of the binding intensity around each peak are shown below. Peaks are sorted by decreasing average binding intensity. (**E**) Cell cycle distribution profiles (propidium iodide staining) of HCT116(WT) and HCT116_RAD21_AID cells in the absence (Control) and presence of auxins (Aux, 12 h of incubation). Representative histograms for one of three biological replicates are shown. (**F**) Growth curves of HCT116 (WT) and HCT116_RAD21_AID cells in the absence and presence of auxins. The average growth curves for four biological replicates are shown on a logarithmic scale on the *y*-axis. Each curve is normalized to the value on the day of auxin addition (day 0). Lines are shown only to connect data points and do not imply linear relationships between measured values. For clarity, the black dotted line marks the relative cell number of 1 across the graph.

**Figure 2 ijms-26-08837-f002:**
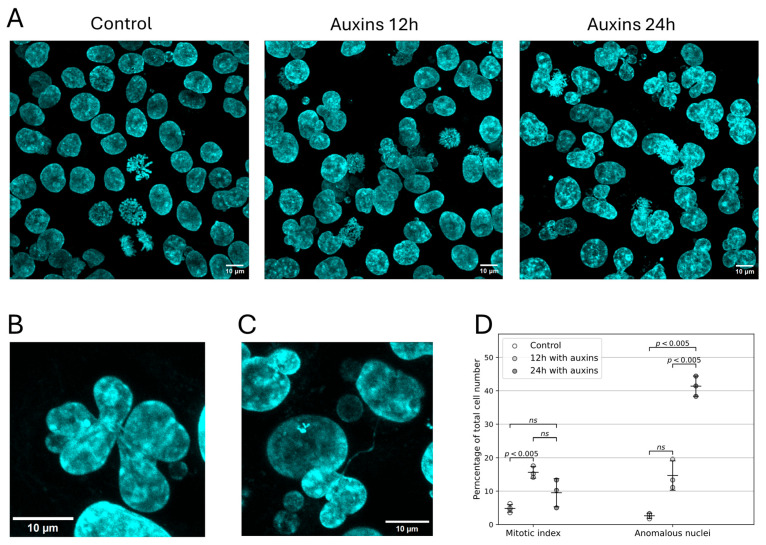
Cohesin depletion increases the mitotic index and leads to the accumulation of cells with abnormal nuclear morphology. (**A**) Hoechst33342-stained nuclei of untreated (Control) and auxin-treated HCT116_RAD21_AID cells for 12 and 24 h. Images correspond to maximum intensity Z-projections of a series of confocal images. (**B**) An example of a nucleus with abnormal (multilobulated) morphology (24 h of cell incubation with auxins). (**C**) An example of nuclei with an internuclear chromatin bridge (24 h of cell incubation with auxins). (**D**) Mitotic index and percentage of cells with abnormal nuclear morphology in untreated cells (Control) and after the indicated incubation times with auxin. Sample sizes (experimental replicates): control, *n* = 4; 12 h with auxins, *n* = 3; 24 h with auxins, *n* = 3. Each replicate included 4–9 fields of view. Black horizontal bars represent the mean of biological replicates in each group; error bars indicate ± standard deviation. Statistical significance was assessed using Welch’s one-way ANOVA followed by the Games-Howell test for multiple comparisons (three pairwise comparisons for each parameter); ns, non-significant difference (*p* > 0.05).

**Figure 3 ijms-26-08837-f003:**
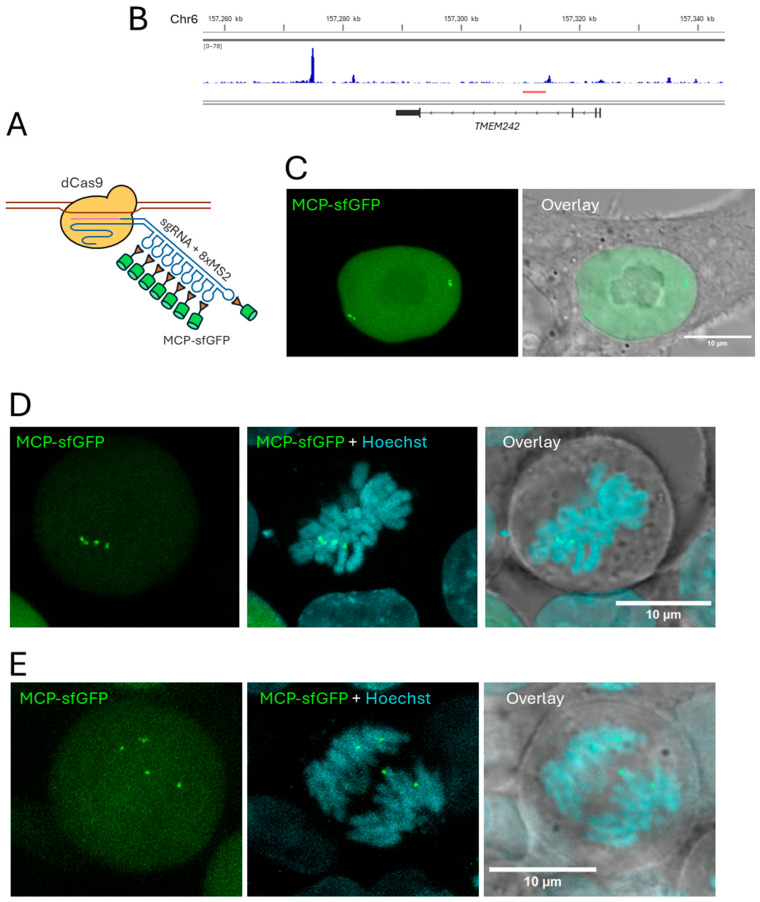
CRISPR-Sirius visualization of the C6 locus in HCT116_RAD21_AID cells. (**A**) CRISPR-Sirius visualization principle (MS2/MCP-sfGFP version). The locus of interest is bound by dCas9 directed by a guide RNA containing eight hairpins of the bacteriophage MS2. The fluorescent protein MCP-sfGFP binds to these hairpins. (**B**) Input-normalized RAD21-binding profile (ChIP-Seq) in the region of the repeat cluster-recognition sites of the guide RNAs, designated as the C6 locus, in HCT116 cells. A red line marks the position of the C6 repeat cluster. The position of the *TMEM242* gene (RefSeq Select annotation) is also marked. The coordinates correspond to the hg38 genome assembly. (**C**) An example of a cell with pairs of replicated C6 loci visualized by CRISPR-Sirius. The GFP channel (maximum intensity Z-projection of confocal images) and an overlay of the GFP and transmitted light channels are shown. (**D**,**E**) Examples of metaphase (**D**) and anaphase (**E**) cells with the C6 locus visualized. The GFP channel (maximum intensity Z-projection of confocal images), an overlay of the GFP and Hoesht33342 channels (maximum intensity Z-projection of confocal images), and an overlay of these channels with transmitted light are shown.

**Figure 4 ijms-26-08837-f004:**
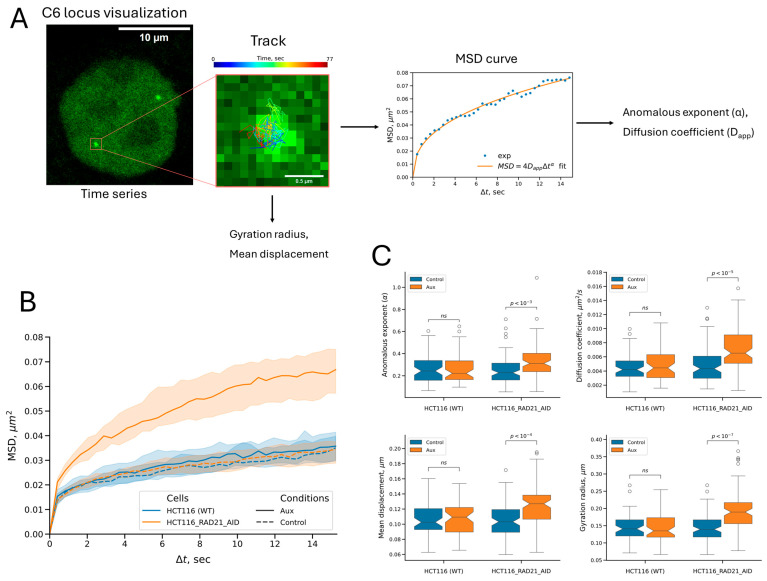
Cohesin depletion increases the mobility of the visualized chromatin locus. (**A**) Workflow of the analysis of the spatial dynamics of the C6 locus. First, a track (a set of signal coordinates in time) was built for the visualized signal using a time-lapse with a time step of 0.412 s. The resulting track was translated into the coordinates of the center of the nucleus to compensate for the mobility of the cell nucleus. The mean displacement per frame and the gyration radius were calculated based on the track coordinates. The tracks were also processed by MSD analysis to obtain the values of the anomalous exponent and the diffusion coefficient. Samples were built from the parameters calculated, which were then processed statistically. (**B**) MSD curves of the C6 locus in control cells without the degron system (HCT116(WT)) and in HCT116_RAD21_AID cells in the absence (Control) and presence of auxins (Aux, 5 h of incubation). The median MSD values for each time lag are shown, with a 95% confidence interval. Sample sizes: HCT116(WT) control, 57 cells; HCT116(WT) + aux, 53 cells; HCT116_RAD21_AID control, 88 cells; HCT116_RAD21_AID + aux, 86 cells. (**C**) Notched boxplots of the C6 locus mobility parameters in cells under the indicated conditions. The position of the median and the 95% confidence interval for the median are indicated by notches. The box indicates the range between the first and third quartiles, and the whiskers mark the position of the point furthest from the median within a 1.5× interquartile range from the edge of the box. The remaining points are marked by circles (outliers). Statistical significance was tested using the Mann–Whitney U-test, with Holm’s correction for multiple comparisons (eight pairwise comparisons in the experiment); ns, non-significant difference.

**Figure 5 ijms-26-08837-f005:**
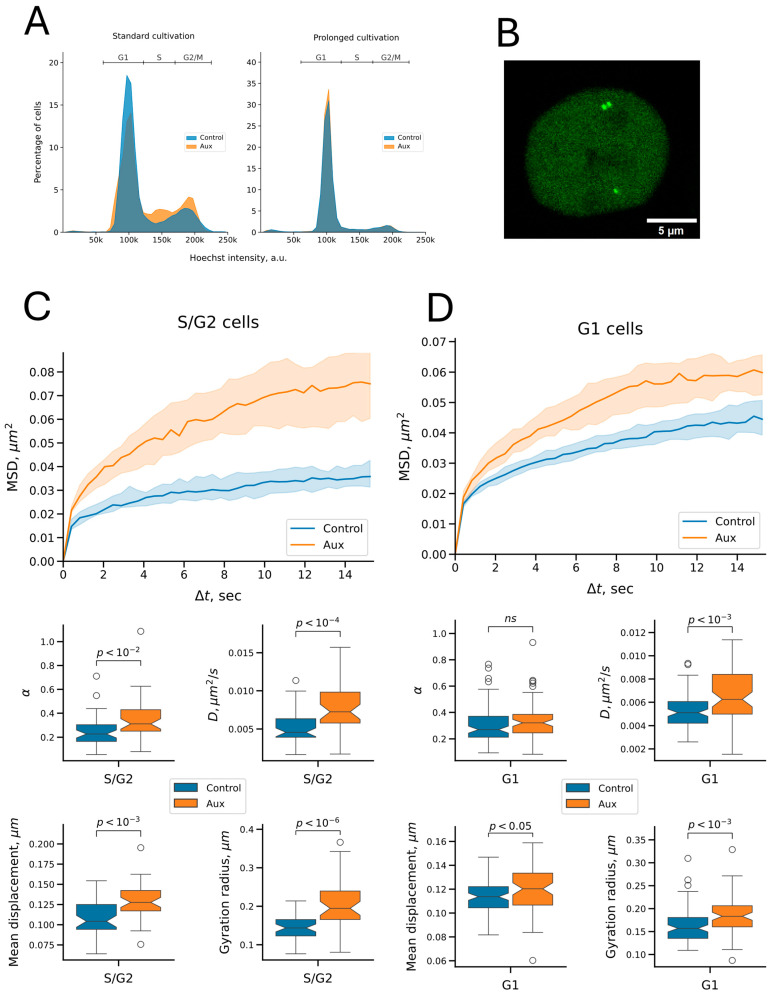
Cohesin restricts the dynamics of both replicated and unreplicated chromatin. (**A**) Cell cycle distribution profiles (Hoechst33342 staining) of HCT116_RAD21_AID cells under standard culture conditions (1 day on a microscopy dish, left) and under prolonged culture conditions (5 days on a microscopy dish, right) that promote the accumulation of cells in the G1 phase of the cell cycle. Representative profiles of three replicates in the absence of auxins (Control) and with the addition of auxins for 5 h (Aux) are shown. (**B**) An example of a cell with a pair of replicated and one presumably unreplicated C6 signal. (**C**) MSD curves and mobility parameters of the C6 locus in HCT116_RAD21_AID cells with a replicated C6 locus (S or G2 phases of the cell cycle) in the absence (Control) and presence of auxins (Aux, 5 h of incubation). Median MSD values for each time lag are shown, with 95% confidence interval values. Sample sizes: control, 44 cells; aux, 41 cells. (**D**) MSD curves and mobility parameters of the C6 locus in HCT116_RAD21_AID cells during long-term cultivation (most cells in the G1 phase of the cell cycle) in the absence (Control) and presence of auxins (Aux, 5 h of incubation). Median MSD values for each time lag are shown, with 95% confidence interval values. Sample sizes: control, 86 cells; aux, 69 cells. In (**C**,**D**), boxplots are shown: boxes represent the interquartile range, whiskers extend to the most distant point within 1.5× the interquartile range, circles indicate outliers, and notches mark the median with its 95% confidence interval. The statistical significance was tested using the Mann–Whitney U test, with Holm’s correction for multiple comparisons (eight pairwise comparisons in the experiment); ns, non-significant difference.

**Figure 6 ijms-26-08837-f006:**
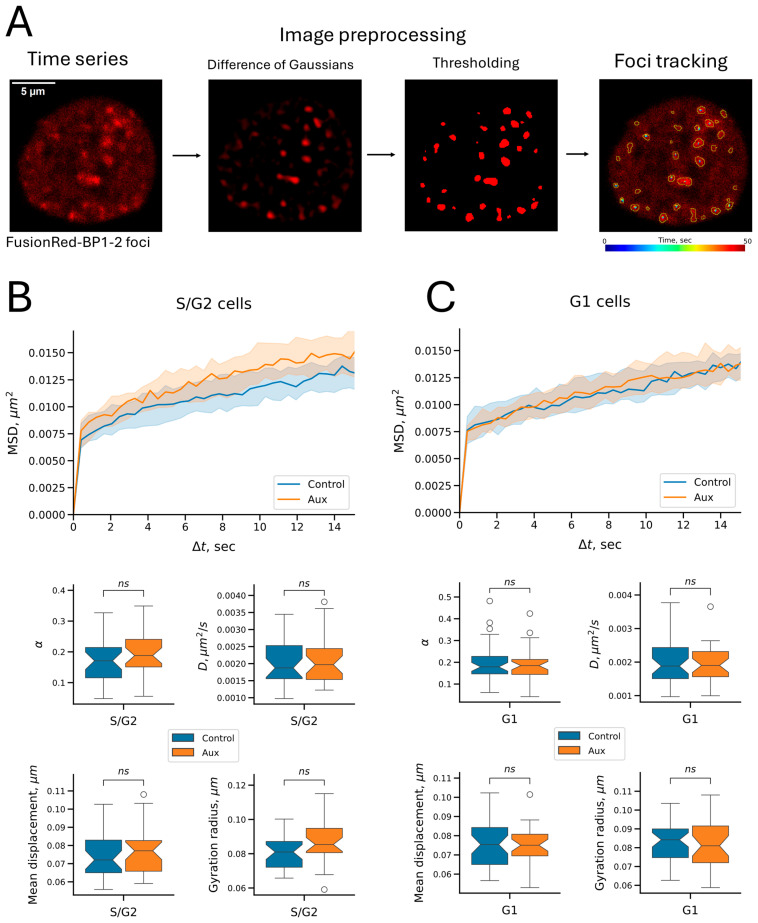
Cohesin does not affect the dynamics of repair foci on a subminute timescale. (**A**) Time-lapse series of images of cells expressing the FusionRed-BP1-2 DSB reporter were recorded to analyze the dynamics of repair foci. Each image was then preprocessed to enhance signal recognition (filtering with the difference in Gaussians and thresholding to convert to a binary image). Repair foci were localized in the images, and the positions of the geometric centers of the foci were determined. The right image shows the trajectories and recognized foci superimposed on the original cell image. Trajectories are shown only for foci that were localized in more than 110 frames. All trajectories were corrected for the movement of the cell nucleus. Motility parameters were calculated based on the corrected trajectories. (**B**) MSD curves and mobility parameters of etoposide-induced FusionRed-BP1-2 foci in HCT116_RAD21_AID cells with a replicated C6 locus (S or G2 phase of the cell cycle) in the absence (control) and presence of auxins (aux, 5 h of preincubation). The median MSD values for each time lag are shown, with 95% confidence intervals. Sample sizes: control, 30 cells; aux, 30 cells. (**C**) MSD curves and mobility parameters of etoposide-induced FusionRed-BP1-2 foci in HCT116_RAD21_AID after long-term culture (most cells in the G1 phase of the cell cycle) in the absence (control) and presence of auxin (aux, 5 h of preincubation). The median MSD values for each time lag are shown, with a 95% confidence interval. Sample sizes: control, 40 cells; aux, 32 cells. In (**B**,**C**), boxplots are shown: boxes represent the interquartile range, whiskers extend to the most distant point within 1.5× the interquartile range, circles indicate outliers, and notches mark the median with its 95% confidence interval. The statistical significance was tested using the Mann–Whitney U test, with Holm’s correction for multiple comparisons (eight pairwise comparisons per experiment); ns, non-significant difference.

**Table 1 ijms-26-08837-t001:** Median values of C6 locus diffusion parameters in replicated and unreplicated chromatin before and after RAD21 depletion.

Parameter	Chromatin State	Untreated Cells	Auxin Treatment	Median Fold Change (Aux/Control)	*p* Adjusted ^1^
Anomalous exponent	unreplicated	0.27	0.32	1.19	1.443 × 10^−1^
	replicated	0.23	0.31	1.37	1.616 × 10^−3^
Diffusion coefficient, μm^2^/s	unreplicated	0.0051	0.0062	1.23	6.470 × 10^−4^
	replicated	0.0046	0.0072	1.59	5.203 × 10^−5^
Mean displacement, μm	unreplicated	0.11	0.12	1.06	1.470 × 10^−2^
	replicated	0.10	0.13	1.22	6.470 × 10^−4^
Gyration radius, μm	unreplicated	0.16	0.18	1.17	6.470 × 10^−4^
	replicated	0.14	0.19	1.35	8.067 × 10^−7^

^1^ The adjusted *p*-values with Holm’s correction for multiple comparisons (eight pairwise comparisons in the experiment) are shown.

**Table 2 ijms-26-08837-t002:** Median values of diffusion parameters of the C6 locus and BP1-2 foci.

Parameter	Chromatin State	Treatment	C6 Locus	BP1-2 Foci	*p*-Value ^1^	*p* Adjusted ^2^
Anomalous exponent	unreplicated	control	0.270573	0.178984	3.551 × 10^−7^	1.065 × 10^−6^
	unreplicated	auxins	0.321870	0.184729	1.863 × 10^−8^	7.453 × 10^−8^
	replicated	control	0.227226	0.170984	4.427 × 10^−3^	4.427 × 10^−3^
	replicated	auxins	0.311817	0.187843	1.321 × 10^−6^	2.641 × 10^−6^
Diffusion coefficient, μm^2^/s	unreplicated	control	0.005100	0.001881	7.646 × 10^−19^	1.147 × 10^−17^
	unreplicated	auxins	0.006248	0.001900	4.930 × 10^−15^	5.916 × 10^−14^
	replicated	control	0.004559	0.001874	7.969 × 10^−11^	4.781 × 10^−10^
	replicated	auxins	0.007236	0.001970	1.104 × 10^−11^	7.729 × 10^−11^
Mean displacement, μm	unreplicated	control	0.113697	0.075425	1.342 × 10^−18^	1.879 × 10^−17^
	unreplicated	auxins	0.120304	0.075046	1.171 × 10^−14^	1.288 × 10^−13^
	replicated	control	0.104229	0.072040	6.333 × 10^−10^	3.167 × 10^−9^
	replicated	auxins	0.127673	0.077045	8.661 × 10^−12^	6.929 × 10^−11^
Gyration radius ^3^, μm	unreplicated	control	0.141783	0.084200	2.831 × 10^−19^	4.529 × 10^−18^
	unreplicated	auxins	0.172551	0.081000	3.094 × 10^−15^	4.022 × 10^−14^
	replicated	control	0.129927	0.081016	4.896 × 10^−12^	4.896 × 10^−11^
	replicated	auxins	0.183876	0.085393	6.255 × 10^−12^	5.629 × 10^−11^

^1^ The *p*-values from the Mann–Whitney U test for each pairwise comparison are shown. ^2^ The adjusted *p*-values with Holm’s correction for multiple comparisons (16 pairwise comparisons in the experiment) are shown. ^3^ The values of the gyration radius for the C6 locus were recalculated for the first 122 frames of the C6 trajectories, corresponding to the length of the FusionRed-BP1-2 tracks.

## Data Availability

ChIP-Seq data have been deposited in the NCBI Gene Expression Omnibus (GEO) with the accession number GSE297215. Raw microscopy images are available from the corresponding author upon request.

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
