# Peer review of "The Influence of Cohesin on the Short-Scale Dynamics of Intact and Damaged Chromatin in Different Phases of the Cell Cycle"

_ijms, 2025, doi:10.3390/ijms26188837_

Round 1
Reviewer 1 Report
Comments and Suggestions for Authors
This manuscript presents a well-designed study investigating the role of cohesin in modulating local chromatin dynamics and the mobility of DNA double-strand break (DSB) repair foci in human cells. The authors utilize a sophisticated experimental system combining auxin-induced depletion of RAD21 (a cohesin subunit) with CRISPR-Sirius live-cell imaging for a specific genomic locus and a FusionRed-BP1-2 reporter for DSB foci. The findings demonstrate that cohesin constrains local chromatin dynamics in both replicated and unreplicated chromatin, with a more pronounced effect in replicated chromatin. Interestingly, cohesin depletion does not affect the mobility of stable DSB repair foci on the studied sub-minute time scales. The paper is generally well-written, and the experiments are meticulously described. The methodologies are robust, and the conclusions are supported by the data.
Specific Comments
- Validation of Auxin-induced Rad21 Degradation:
- While auxin-induced RAD21 degradation is a well-established method, the authors' detailed validation of their specific HCT116_RAD21_AID cell line is crucial. However, placing all validation results (especially the kinetics of depletion and ChIP-Seq profiles) in the main Figure 1 might be a bit extensive. It would be more streamlined to present the primary evidence for functional depletion (e.g., cell cycle block and proliferation curve) in Figure 1, and move the detailed kinetics (Western blot and ChIP-Seq profiles) to a Supplementary Figure. This would allow the main figure to focus on the functional consequences directly relevant to the study's core.
- Crucially, Figure 1B (Western blot) lacks a loading control. This is a standard requirement for Western blot analysis to ensure equal protein loading across lanes. Please add a loading control (e.g., β-actin, GAPDH, or total protein staining like Ponceau S as mentioned in the methods but not shown in the figure) to Figure 1B or the corresponding Supplementary Figure.
- Transcriptional Activity of the TMEM242 Gene:
- The authors selected the C6 locus within the TMEM242 gene for visualization. It would be highly beneficial to know the transcriptional status of this gene in HCT116 cells. Is TMEM242 an actively transcribed gene, or is it largely inactive?
- If TMEM242 is actively transcribed, does cohesin depletion alter its expression level? Given cohesin's role in gene regulation and chromatin architecture, understanding any transcriptional changes at this specific locus after RAD21 depletion would provide valuable context for interpreting the observed mobility changes. Please discuss this point and, if possible, provide supporting data (e.g., RNA-Seq or RT-qPCR for TMEM242 expression with/without auxin treatment).
- Impact of Transcription on Chromatin Mobility:
- The manuscript focuses on one specific locus. Chromatin mobility is known to be influenced by various factors, including transcriptional activity, transcription factor binding, and overall chromatin structure. It would significantly strengthen the biological breadth and impact of this study if the authors could explore and discuss how transcription generally affects chromatin mobility in their system, both with and without the cohesin complex.
- For instance, is there a detectable difference in local chromatin mobility between actively transcribed and inactive genes? Could the authors select a known actively transcribed gene and a known repressed gene and measure their respective mobilities using the established CRISPR-Sirius system? This comparative analysis would provide valuable insights into the interplay between cohesin, transcription, and chromatin dynamics. While adding new experimental data might be beyond the scope of a revision, a thorough discussion of this aspect, referencing existing literature where appropriate, would be a great addition to the discussion section.
- Typographical Correction:
- Line 211: "Yates' correction" should be corrected to "Yates's correction."
Overall Recommendation
This is a high-quality manuscript that addresses an important question in chromatin biology and DNA repair. The experimental design is sound, the data are generally well-presented and analyzed, and the conclusions are supported by the data. I recommend this paper for publication after addressing the specific comments mentioned above, particularly regarding the Western blot loading control, the transcriptional status of the visualized locus, and a more comprehensive discussion of transcriptional effects on chromatin mobility.
Author Response
Comments 1: "This manuscript presents a well-designed study investigating the role of cohesin in modulating local chromatin dynamics and the mobility of DNA double-strand break (DSB) repair foci in human cells. The authors utilize a sophisticated experimental system combining auxin-induced depletion of RAD21 (a cohesin subunit) with CRISPR-Sirius live-cell imaging for a specific genomic locus and a FusionRed-BP1-2 reporter for DSB foci. The findings demonstrate that cohesin constrains local chromatin dynamics in both replicated and unreplicated chromatin, with a more pronounced effect in replicated chromatin. Interestingly, cohesin depletion does not affect the mobility of stable DSB repair foci on the studied sub-minute time scales. The paper is generally well-written, and the experiments are meticulously described. The methodologies are robust, and the conclusions are supported by the data."
Response 1: We are thankful for the thorough review of our manuscript.
Comments 2: "Validation of Auxin-induced Rad21 Degradation:
While auxin-induced RAD21 degradation is a well-established method, the authors' detailed validation of their specific HCT116_RAD21_AID cell line is crucial. However, placing all validation results (especially the kinetics of depletion and ChIP-Seq profiles) in the main Figure 1 might be a bit extensive. It would be more streamlined to present the primary evidence for functional depletion (e.g., cell cycle block and proliferation curve) in Figure 1, and move the detailed kinetics (Western blot and ChIP-Seq profiles) to a Supplementary Figure. This would allow the main figure to focus on the functional consequences directly relevant to the study's core."
Response 2: Indeed, Figure 1 may appear somewhat overloaded; however, we consider it important to present all aspects of model validation in a single figure. The data shown, particularly the depletion curve, provide the rationale for selecting 5 hours of auxin incubation as the time point for microscopic examination of the cells and for assessing the effects of depletion. Readers who are not interested in the detailed validation of the model may skip this section and proceed directly to the subsequent parts of the paper describing the experiments.
Comments 3: "Crucially, Figure 1B (Western blot) lacks a loading control. This is a standard requirement for Western blot analysis to ensure equal protein loading across lanes. Please add a loading control (e.g., β-actin, GAPDH, or total protein staining like Ponceau S as mentioned in the methods but not shown in the figure) to Figure 1B or the corresponding Supplementary Figure."
Response 3: The loading control (Ponceau S staining of total protein) for the Western blot shown in Figure 1B is provided in Supplementary Figure S2. Supplementary Figure S2 also includes the uncropped image of the ECL-developed membrane shown in Figure 1B, as well as two additional experimental replicates. Because Figure 1 is already quite dense, we chose to present the loading control in the supplement. We have now added a reference to Supplementary Figure S2 in the legend to Figure 1B (lines 173-174).
Comments 4: "Transcriptional Activity of the TMEM242 Gene:
The authors selected the C6 locus within the TMEM242 gene for visualization. It would be highly beneficial to know the transcriptional status of this gene in HCT116 cells. Is TMEM242 an actively transcribed gene, or is it largely inactive?
If TMEM242 is actively transcribed, does cohesin depletion alter its expression level? Given cohesin's role in gene regulation and chromatin architecture, understanding any transcriptional changes at this specific locus after RAD21 depletion would provide valuable context for interpreting the observed mobility changes. Please discuss this point and, if possible, provide supporting data (e.g., RNA-Seq or RT-qPCR for TMEM242 expression with/without auxin treatment)."
Response 4: Indeed, adding information on TMEM242 gene expression would provide a better context for the dynamics under study. TMEM242 encodes a mitochondrial transmembrane protein involved in the assembly of ATP synthase (Carroll et al., 2021). To address the question regarding the expression level of this gene and the effect of cohesin depletion on its expression, we examined the PRO-Seq data from (Rao et al., 2017) on HCT116 cells with an auxin degron system similar to our cells (data are available in the GEO database under accession number GSE106886). The PRO-Seq method allows analysis of not the total transcriptome, but only the transcripts that are currently synthesized (the so-called nascent transcripts). According to these data, TMEM242 is expressed at a very low level in HCT116 cells. Specifically, the average transcripts per million (TPM) for this gene in untreated cells was 14.5, which is much lower than, for example, the average TPM for the housekeeping gene GAPDH in the same cells (TPM = 1106.4). Under these circumstances, it appears unlikely that cohesin depletion would have a significant effect on the expression of this gene. Indeed, based on the PRO-Seq data described by Rao et al. (2017), no change in the number of nascent TMEM242 transcripts is observed upon cohesin depletion (6 hours of incubation with auxin) (log2FoldChange = -0.011176, adjusted p-value = 0.927, according to DESeq2 analysis).
It should be noted that using a method capable of quantifying nascent transcripts (e.g., PRO-Seq) is essential, as the overall transcript level for the target gene is unlikely to change over a short incubation period with auxins (a few hours). The described data have been added to the Results section of the manuscript (lines 230-235), and the analysis of these data is also detailed in the Methods section (lines 931-939).
References:
Carroll, J.; He, J.; Ding, S.; Fearnley, I.M.; Walker, J.E. TMEM70 and TMEM242 Help to Assemble the Rotor Ring of Human ATP Synthase and Interact with Assembly Factors for Complex I. Proc. Natl. Acad. Sci. U.S.A. 2021, 118, e2100558118, doi:10.1073/pnas.2100558118.
Rao, S.S.P.; Huang, S.-C.; Glenn St Hilaire, B.; Engreitz, J.M.; Perez, E.M.; Kieffer-Kwon, K.-R.; Sanborn, A.L.; Johnstone, S.E.; Bascom, G.D.; Bochkov, I.D.; et al. Cohesin Loss Eliminates All Loop Domains. Cell 2017, 171, 305-320.e24, doi:10.1016/j.cell.2017.09.026.
Comments 5: "Impact of Transcription on Chromatin Mobility:
The manuscript focuses on one specific locus. Chromatin mobility is known to be influenced by various factors, including transcriptional activity, transcription factor binding, and overall chromatin structure. It would significantly strengthen the biological breadth and impact of this study if the authors could explore and discuss how transcription generally affects chromatin mobility in their system, both with and without the cohesin complex.
For instance, is there a detectable difference in local chromatin mobility between actively transcribed and inactive genes? Could the authors select a known actively transcribed gene and a known repressed gene and measure their respective mobilities using the established CRISPR-Sirius system? This comparative analysis would provide valuable insights into the interplay between cohesin, transcription, and chromatin dynamics. While adding new experimental data might be beyond the scope of a revision, a thorough discussion of this aspect, referencing existing literature where appropriate, would be a great addition to the discussion section."
Response 5: These are interesting questions about how transcription affects gene mobility and whether cohesin depletion has a differential effect on gene mobility depending on expression level. We have reviewed and summarized several recent papers on the relationship between transcription and mobility in the revised Discussion section (lines 572-608). To our knowledge, the second question (regarding a possible differential effect of cohesin depletion) has not yet been addressed, and we agree that it would be highly valuable to investigate this in the future. However, such an analysis would require considerably more work than was feasible for the current study. This is mainly due to the fact that the efficiency of locus visualization using the CRISPR-Sirius system (as well as other systems, in our experience) is far from 100%. Most loci we attempted to visualize could either not be detected at all, or were visualized with low efficiency, as we discussed in our previous work (Viushkov et al., 2024). To collect sufficient samples of both active and inactive loci for a robust comparison would require testing a large number of targets. In our view, such work goes beyond the scope of the present study, which is focused on a detailed analysis of a single model locus. Nevertheless, we believe that this line of research is promising and could be pursued in the future, especially if visualization efficiency of the CRISPR-Sirius system can be improved or more loci suitable for visualization become available.
References:
Viushkov, V.S.; Lomov, N.A.; Rubtsov, M.A. A Comparison of Two Versions of the CRISPR-Sirius System for the Live-Cell Visualization of the Borders of Topologically Associating Domains. Cells 2024, 13, 1440, doi:10.3390/cells13171440.
Comments 6: "Typographical Correction: Line 211: "Yates' correction" should be corrected to "Yates's correction." "
Response 6: Thank you for pointing this out. However, following the suggestions of the second reviewer, we revised the analysis of this experiment and implemented a more appropriate statistical test. In the revised analysis, Yates’s correction is no longer applied.
Comments 7: "Overall Recommendation. This is a high-quality manuscript that addresses an important question in chromatin biology and DNA repair. The experimental design is sound, the data are generally well-presented and analyzed, and the conclusions are supported by the data. I recommend this paper for publication after addressing the specific comments mentioned above, particularly regarding the Western blot loading control, the transcriptional status of the visualized locus, and a more comprehensive discussion of transcriptional effects on chromatin mobility."
Response 7: Thank you for your valuable comments and ideas for further development of this work. The effect of transcription on mobility in case of cohesin depletion would be indeed interesting to study in the future.
Reviewer 2 Report
Comments and Suggestions for Authors
The article of Viushkov et al. "The influence of cohesin on the short-scale dynamics of intact and damaged chromatin in different phases of the cell cycle" describes the role of cohesinin the regulation of the dynamics of damaged chromatin. The authors investigated the effect of cohesin depletion on the spatial dynamics of a genomic locus under consideration of the influence on replicated and not replicated chromatin. They generated HCT116 cells with a system for auxin-induced depletion of the RAD21 subunit of the cohesin complex and used the CRISPR-Sirius system for chromatin visualization. By means of confocal time-lapse microscopy and elucidation of the biophysical parameters of the trajectories, the authors showed that RAD21 subunit depletion increased the mobility of the visualized genomic locus in replicated and unreplicated chromatin. The authors also investigated the effect of cohesin depletion on the spatial dynamics of repair foci after induction of double-strand breaks by the etoposide. Changes in the mobility of the BP1-2 foci upon cohesin depletion, either in G1 cells or S/G2 cells were not detected suggesting that repair foci are fairly stable structures which spatial stability does not depend on cohesin.
The article is well written and can be accepted after minor revision:
1.) Figure 1B: The curve looks strange with vertices at the really measure values. In addition they suggest results just by linear connection of neighbouring results. For instance at the the time value 7 h the band intensity (not measured) is the mean between 6 and 8 hours. Moreover the error ranges look strange. I recoomend to exchange this graph by a bar histogram for the values really measured.
2.) The same as in 1.) is recommended for Figure 1F
3.) exchange yH2AX by γH2AX throughout the text
4.) In the legend of Figure 2: Are the number of cells randomly chosen from the replicates or are these the number of cells for each replicate. Since you finally evaluate single cell results, how dis you determine that the quality of the replicates was equal. Otherwise you would bias the results if you use more or less cells from a more or less better replicate.
5.) Materials and Methods: Please, add the numerical aperture (NA) of the microscope objective since this determines the optical resolution. Otherwise all other data of resolution/resolution improvement cannot be interpreted.
Author Response
Comments 1: "The article of Viushkov et al. "The influence of cohesin on the short-scale dynamics of intact and damaged chromatin in different phases of the cell cycle" describes the role of cohesinin the regulation of the dynamics of damaged chromatin. The authors investigated the effect of cohesin depletion on the spatial dynamics of a genomic locus under consideration of the influence on replicated and not replicated chromatin. They generated HCT116 cells with a system for auxin-induced depletion of the RAD21 subunit of the cohesin complex and used the CRISPR-Sirius system for chromatin visualization. By means of confocal time-lapse microscopy and elucidation of the biophysical parameters of the trajectories, the authors showed that RAD21 subunit depletion increased the mobility of the visualized genomic locus in replicated and unreplicated chromatin. The authors also investigated the effect of cohesin depletion on the spatial dynamics of repair foci after induction of double-strand breaks by the etoposide. Changes in the mobility of the BP1-2 foci upon cohesin depletion, either in G1 cells or S/G2 cells were not detected suggesting that repair foci are fairly stable structures which spatial stability does not depend on cohesin. "
Response 1: Thank you for the thorough review of our manuscript.
Comments 2: "1). Figure 1B: The curve looks strange with vertices at the really measure values. In addition they suggest results just by linear connection of neighbouring results. For instance at the the time value 7 h the band intensity (not measured) is the mean between 6 and 8 hours. Moreover the error ranges look strange. I recoomend to exchange this graph by a bar histogram for the values really measured. "
Response 2: We agree. For clarity, we have plotted the actual measured experimental values for each replicate and removed the lines connecting them. The figure legend has been modified accordingly (lines 170, 171–173).
Comments 3: "2). The same as in 1.) is recommended for Figure 1F "
Response 3: This plot (as well as the original Figure 1B) was drawn using the lineplot function from the Seaborn scientific visualization library (https://seaborn.pydata.org/generated/seaborn.lineplot.html). For example, this type of plot was used in a similar work to ours on auxin degron cells, also to display growth curves (see Figure S6A in the Supporting Information of Gabrielle et al., 2022, and Figure S5A in the same paper). This approach has become increasingly popular, as it provides a clear representation of error bands without cluttering the figure with an excessive number of whiskers. Moreover, when data points are close to each other (as in our curves for control cells and untreated cells with the degron system), such error bands facilitate correct interpretation by showing more clearly which error bars correspond to which curve.
We attempted to redraw the plot according to your recommendations, but in our opinion the resulting figure was less clear, primarily due to the crossing of whiskers. Therefore, we would prefer to keep the original representation. We would also like to retain the connecting lines in the growth curves, as these lines help to trace the relationship between points within the same experimental condition. To avoid misinterpretation, we have added a note to the figure caption stating that the lines are shown for illustrative purposes only and do not imply linear changes between experimental points (lines 186-187).
References:
Gabriele, M.; Brandão, H.B.; Grosse-Holz, S.; Jha, A.; Dailey, G.M.; Cattoglio, C.; Hsieh, T.-H.S.; Mirny, L.; Zechner, C.; Hansen, A.S. Dynamics of CTCF- and Cohesin-Mediated Chromatin Looping Revealed by Live-Cell Imaging. Science 2022, 376, 496–501, doi:10.1126/science.abn6583.
Comments 4: "3.) exchange yH2AX by γH2AX throughout the text"
Response 4: Thank you for pointing this out, we have corrected the typo (line 79).
Comments 5: "4.) In the legend of Figure 2: Are the number of cells randomly chosen from the replicates or are these the number of cells for each replicate. Since you finally evaluate single cell results, how dis you determine that the quality of the replicates was equal. Otherwise you would bias the results if you use more or less cells from a more or less better replicate."
Response 5: Thank you for your comment. It prompted us to reconsider our approach to analyzing this experiment. The numbers in the legend to the original Figure 2 correspond to the total number of cells in each condition. For each condition (control, 12 h incubation, 24 h incubation), in our original analysis, we pooled replicates by summing the counts of all event types (mitoses, abnormal nuclei, total cells). In this pooling, we included all available events from each replicate without altering their numbers, avoiding any manipulative weighting. Although replicate sizes within each experimental condition were comparable, our original approach was, in fact, pseudoreplication (Lazic et al., 2018; Lord et al., 2020), since cells or even fields of view on the same microscopy dish are not independent observations. This may lead to artificially increased statistical power and a higher likelihood of type I errors.
A more correct approach is to calculate the average frequency of each cell type (mitoses, abnormal nuclei) per dish and treat the microscopy dishes as biological replicates (Lazic et al., 2018; Lord et al., 2022). Frequencies of each event type can then be compared between experimental conditions using Welch's version of one-way ANOVA followed by Games-Howell test, a variation of Tukey’s honestly significant difference (HSD) test that does not assume equal variance of samples. This method is more conservative than our original approach and addresses the concern you raised. While this results in a predictable decrease in statistical power, the main trends are preserved: an increase in the proportion of mitotic cells with cohesin depletion at 12 h, and accumulation of abnormal nuclei by 24 h of auxin treatment. We have modified Figure 2D and its legend accordingly to reflect this updated analysis (lines 211-216).
References:
Lazic SE, Clarke-Williams CJ, Munafò MR (2018) What exactly is ‘N’ in cell culture and animal experiments? PLoS Biol 16(4): e2005282. https://doi.org/10.1371/journal.pbio.2005282
Samuel J. Lord, Katrina B. Velle, R. Dyche Mullins, Lillian K. Fritz-Laylin; SuperPlots: Communicating reproducibility and variability in cell biology. J Cell Biol 1 June 2020; 219 (6): e202001064. doi: https://doi.org/10.1083/jcb.202001064
Comments 6: "5.) Materials and Methods: Please, add the numerical aperture (NA) of the microscope objective since this determines the optical resolution. Otherwise all other data of resolution/resolution improvement cannot be interpreted."
Response 6: We agree with your comment and have included the numerical aperture value (1.42) in the Materials and Methods section (lines 809, 950 and 1047).